# Engineering intercellular communication using M13 phagemid and CRISPR-based gene regulation for multicellular computing in *Escherichia coli*

Hadiastri Kusumawardhani ✉, Florian Zoppi, Roberto Avendaño & Yolanda Schaerli ✉

Engineering multicellular consortia, where information processing is distributed across specialized cell types, offers a promising strategy for implementing sophisticated biocomputing systems. However, a major challenge remains in establishing orthogonal intercellular communication, or "wires," within synthetic bacterial consortia. In this study, we address this bottleneck by integrating phagemid-mediated intercellular communication with CRISPR-based gene regulation for multicellular computing in synthetic *E. coli* consortia. We achieve intercellular communication with high sensitivity by regulating the transfer of single guide RNAs (sgRNAs) encoded on M13 phagemids from sender to receiver cells. Once inside the receiver cells, the transferred sgRNAs mediate gene regulation via CRISPR interference. Leveraging this approach, we successfully constructed one-, two-, and four-input logic gates. Our work expands the toolkit for intercellular communication and paves the way for complex information processing in synthetic microbial consortia, with diverse potential applications, including biocomputing, biosensing, and biomanufacturing.

Synthetic biology provides new solutions to agriculture, healthcare, manufacturing, and environmental challenges[1–4]. An important goal of the field is to engineer genetic circuits that process information to produce programmable outputs, essentially creating biologically-based computing devices[5–8]. Inspired by electronic computing, Boolean logic gates have been extensively implemented in living cells[7–10]. The resulting organisms have a wide range of applications. For example, they have been designed for environmental biosensing, such as detecting heavy metals[11]. In plants, engineered logic gates integrated multiple stress signals to trigger a defence response[12]. Additionally, using logic gates, therapeutic bacteria were targeted to specific microenvironments like the gastrointestinal tract and tumours[13]. However, load, burden, genetic crosstalk, retroactivity,

stoichiometric mismatch, and the limited amount of well-characterized transcriptional units limit the computational capabilities that can be implemented[14–17].

Traditionally, most synthetic circuits have employed protein transcription factors to regulate gene expression. However, CRISPR (clustered regularly interspaced short palindromic repeats)-based genetic regulation presents a promising alternative, addressing many challenges associated with synthetic circuit construction. This approach, known as CRISPR interference (CRISPRi) when used for transcriptional repression, involves a single guide RNA (sgRNA) that guides a catalytically inactive form of Cas9 (dCas9) to promoter or coding regions, effectively blocking transcription[18,19]. CRISPR-based gene expression regulation offers several advantages, including

Department of Fundamental Microbiology, Faculty of Biology and Medicine, University of Lausanne, Lausanne, Switzerland.
✉e-mail: hadiastri.kusumawardhani@unil.ch; yolanda.schaerli@unil.ch

reduced crosstalk between components due to highly specific RNA–DNA interactions, a lower metabolic burden from decreased protein production, and the straightforward design of orthogonal variants[20–22].

Another promising route to mitigate problems in constructing synthetic circuits is the engineering of multicellular consortia where the computation is distributed across different specialized cell types. Multicellular computing holds the potential to perform sophisticated biological computation that is too complex to be implemented in monocultures offering enhanced robustness, modularity, scalability, and component reusability[14,23,24]. The challenge of expanding the genetic circuitry into multicellular consortia lies in establishing a synthetic intercellular communication or "wiring" system. Impressive efforts have been made to implement quorum-sensing molecules and other small molecules as intercellular communication systems[24–27]. However, each of these small molecule wires is only capable of sending a single message, while the number of truly orthogonal 'wires' is still limited. In contrast, in electronic computers, wiring is established with one-on-one connections using well-insulated conductive wires. The same type of wires can be used for all connections and each wire is capable of transmitting different messages in a standard format. Reflecting on this system, ideally, multicellular consortia would use an intercellular communication system with one sort of wires that can be used to easily send and receive different signals simultaneously. The exchange of genetic material via bacterial conjugation[28–30] or phages[31] has been proposed to constitute such wires. Marken and Murray developed an addressable and adaptable framework for DNA-based communication in *Escherichia coli*, by leveraging plasmid conjugation[29]. In a separate study, Weiyue et al. employed bacterial conjugation to deliver an inducible CRISPRi system, repressing the mRFP gene in a target *E. coli* strain[30]. Ortiz and Endy engineered cell-cell communication within a multicellular consortium of *E. coli* using M13 phage[31]. They employed the M13 phage to transfer the genetic information for T7 RNA polymerase from sender to receiver cells, where it subsequently activated reporter gene expression. M13 phage is well suited for synthetic intercellular communication because it is highly efficient in transferring non-M13 genetic material on phagemids —plasmids containing the M13 phage DNA packaging sequence[31,32]. Moreover, its release is not detrimental to the host cell because it is a non-lytic phage[33].

Here, we combine CRISPR-based gene regulation with phagemid-mediated intercellular communication for multicellular computing in *E. coli* (Fig. 1). Specifically, we engineered M13 phages to transfer sgRNAs on phagemids between co-cultured sender and receiver cell populations. In combination with the dCas9 expressed in the receiver cells, the transduced sgRNA inhibits transcription of a reporter gene directly or via a CRISPRi-cascade in the receiver cell population. Moreover, we can induce or block the transfer of the phagemid from multiple sender strains with chemical inducers. Using this approach, we have successfully built one-, two- and four-input logic gates where the complex genetic circuitries are divided into members of a consortium of *E. coli* strains. A direct comparison with quorum-sensing-based communication revealed that phagemid-based communication exhibits higher sensitivity. Our work expands the toolkit for intercellular communication and establishes a foundation for complex information processing in synthetic consortia.

## Results
### Establishing a M13 phagemid system for intercellular communication

For gene regulation in our engineered microbial consortium of *E. coli* strains we employ CRISPRi[21,34]. To establish communication between consortium members, we combined CRISPRi with a M13 phagemid system[33]. The phagemid encodes the message in form of a DNA sequence and it is shuttled from sender to receiver cells by M13 phages. The release and the absorption of M13 phages requires the expression of F pili from the host *E. coli* cells, thus we chose to work with JM101 strain - one of the original strains described for M13 phage propagation[35,36].

We first built two types of message phagemids. The first one is derived from a pBR322-based plasmid backbone (originated from pET) and the second from a RSF1030-based plasmid backbone (Fig. 2A). In addition to the *E. coli* origin of replication and an antibiotic resistance cassette (ampicillin resistance for pBR322-based plasmid and gentamicin resistance for RSF1030-based plasmid), they contain the M13 packaging signal (F1 ori) and the message to be sent: a sgRNA downstream of a constitutive promoter J23110. The message phagemids were transformed into sender cells that also contained a helper plasmid (HP17_KO7)[37]. It encodes for the phage gene cluster to create functional M13 virions, but it does not contain a phage packaging

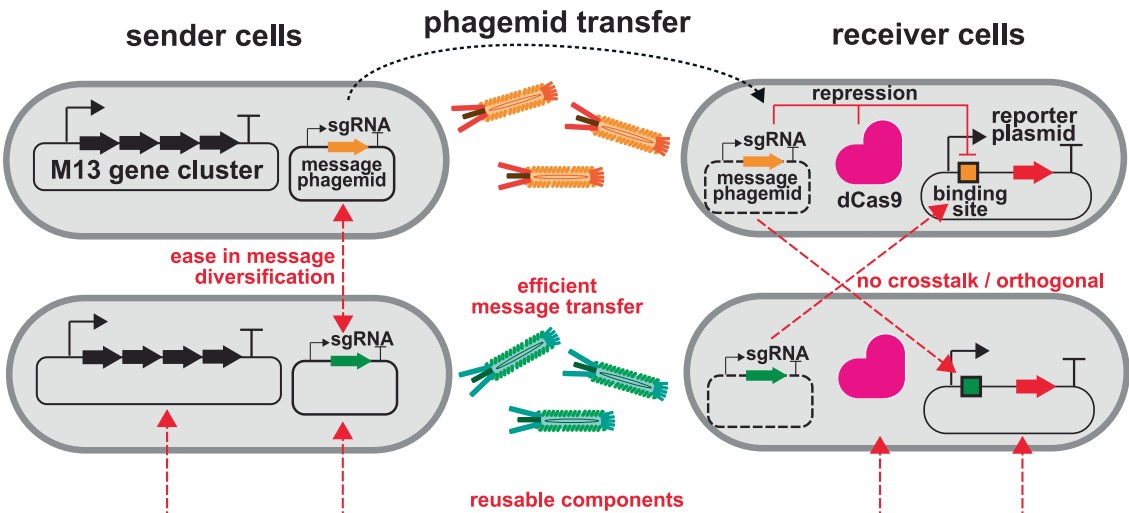

**Fig. 1 | Design of intercellular communication using M13 phage and CRISPR-based gene regulation.** Intercellular communication is achieved through the exchange of genetic material between sender and receiver cells mediated by M13 phage transfer. M13 phage particles contain a plasmid with a M13 packaging signal (phagemid) that encodes the DNA message, in this case a single guide RNA. This communication system allows for reusable components, ease in message diversification (i.e., exchanging sgRNA with another DNA message), efficient message transfer (>97% transfer within 4 h of co-culturing sender and receiver cells), and orthogonality between different sgRNA messages.

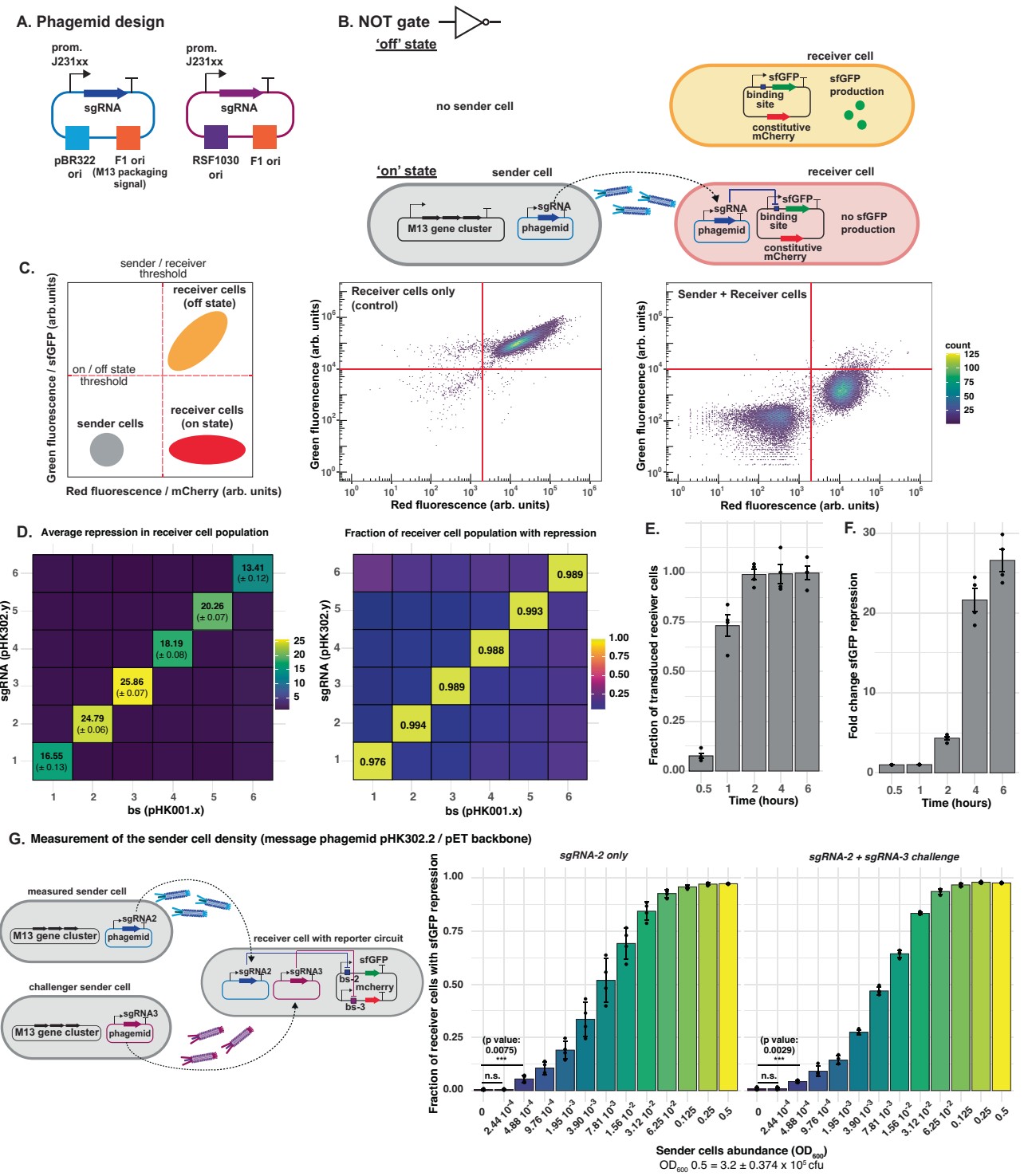

**A. Phagemid design**

**B. NOT gate**

**C.**

**D.** Average repression in receiver cell population

Fraction of receiver cell population with repression

**E.**

**F.**

**G.** Measurement of the sender cell density (message phagemid pHK302.2 / pET backbone)

signal and it is therefore not transduced. The receiver cells carried a plasmid coding for dCas9 and csy4 nuclease (pJ1996v2)[21] and a reporter plasmid. Csy4, an RNase from the CRISPR system of *Pseudomonas aeruginosa*[38], was used to prevent mRNA context-dependency and ensure transcriptional insulation. A 20 bp csy4 cleavage sequence flanks the sgRNAs and is also placed upstream of reporter genes[21]. The reporter plasmid contains a sgRNA-binding site and a super folder green fluorescence protein (sfGFP) reporter gene, both placed downstream of a J23100 promoter, so that binding of the dCas9-sgRNA complex can inhibit transcription of sfGFP. The receiver cells also constitutively express mCherry from the reporter plasmid to facilitate quantification of transduction.

For the transfer of the pET-based message phagemid from sender to receiver cells, we co-cultured them at an initial ratio of 1:1 in liquid 2x Yeast Extract Tryptone (2x YT) medium at 37 °C for 4 h in the absence of any antibiotic selection. In combination with dCas9 expressed in the receiver cells, the transmitted sgRNA inhibited transcription of sfGFP, thus creating an inverter/'NOT' gate (Fig. 2B). This repression of sfGFP was measured using flow cytometry (Fig. 2C). Depending on the sgRNA used, we observed 13–25-fold sfGFP repression with the pBR322-based phagemid and a J23110 promoter upstream of the sgRNA and more than 97% of the receiver cells showed repressed reporter levels (Fig. 2D). Changing the promoter upstream of sgRNA to a stronger promoter (i.e., J23119) increased sfGFP repression up to 60-fold

**Fig. 2 | Characterization of phagemid transfer. A** Schematic representations of the two types of phagemids. **B** An inverter/'NOT' gate design with constitutive phagemid transfer between sender and receiver cells. The phagemid coding for a sgRNA is packaged into M13 phage particles by sender cells and transduced to receiver cells. Together with dCas9 (not shown) in the receiver cells sgRNA inhibits reporter expression. **C** Measurement of phagemid transfer between sender and receiver cells with flow cytometry. Schematic of where sender and receiver cells are expected in a flow cytometry plot where sfGFP expression is plotted against mCherry fluorescence (left). Flow cytometry measurement of sfGFP and mCherry reporters of receiver cells only (middle) or upon co-culturing sender and receiver cells at 37 °C for 4 h (right). **D** Orthogonality assay of different sgRNAs in sender pBR322-based phagemids (pHK302.y, ampicillin resistance) (y axis) and binding sites in receiver cells (pHK001.x) (x axis). The initial sender-to-receiver ratio was 1:1. Left: Mean ± SD of 3 independent biological replicate of sfGFP repression fold-change determined by flow cytometry. Right: fraction of receiver cells with fluorescent below the chosen threshold (shown in **C**). Data represent the mean of 3 independent biological replicates. **E** Transduction of pBR322-based message phagemid (with gentamicin resistance) over time determined by calculating colony-forming units of successfully transduced cells using selective plating divided by total amount of receiver colonies. The initial sender-to-receiver ratio was 2:1. Corresponding data of pBR322 with ampicillin resistance and for the RSF1030 backbone can be found in Fig. S1. **F** Fold-change of sfGFP reporter repression over time for pBR322-based phagemid. Data in E and F represent the mean ± SD of 4 independent biological replicates. **G** Measurement of the sender cell density using pET/pBR322-based phagemids. Receiver cells harbored two independent NOT gates allowing the detection of two distinct sender cell populations independently. The sensitivity of this circuit was tested by co-culturing receiver cells at $OD_{600}$ 0.5 ($3.2 \pm 0.375 \times 10^5$ cfu) with only sender cells sending sgRNA-2 at different densities or challenged by an additional sender cell population at $OD_{600}$ 0.5 sending sgRNA-3. Data in G represent the mean ± SD of 4 independent biological replicates. Student's t-test was performed to determine statistical significance between the samples with and without sender cells added. Source data are provided as a Source Data file.

(Fig. S1C). To test for orthogonality, we built the NOT gate with six different sgRNAs on the message phagemid and the corresponding binding sites on the reporter plasmid (Fig. 2D). Only the matching sgRNA and binding site pairs repressed sfGFP expression.

Next, we quantified the dynamics of phagemid transduction between sender and receiver cells. Using a 2:1 sender-to-receiver cell ratio (see justification in the next paragraph), we took samples at 0.5, 1, 2, 4, and 6 h. We then plated the cultures on two types of agar plates: one with antibiotics allowing growth of all receiver cells and another that selectively allowed growth of only those with the phagemid. We counted the colony-forming units on each plate. In this experiment, both message phagemids contained a gentamicin resistance gene to ensure comparability. We observed phagemid transduction occurring within 2 h for both pBR322 and RSF1030 phagemids (Figs. 2E and S1A). Additionally, we also followed sfGFP repression at these timepoints. It took around 4 h to record a 20-fold sfGFP repression for both constructs (Figs. 2F and S1B).

Furthermore, we also compared the effect of different antibiotic pressures on the pBR322 phagemid, namely gentamicin versus ampicillin (Fig. S1C, D). To achieve a level of repression comparable to that of the ampicillin-selected phagemid with a 1:1 sender-to-receiver cell ratio, the ratio had to be increased to 2:1 for the gentamicin-selected phagemid. Even with this adjustment, the number of colonies growing after transduction during the first hour was lower with the gentamicin-selected phagemid (Fig. S1C), while the sfGFP- fold repression was comparable to that of the ampicillin-selected phagemid (Fig. S1D). We attribute these differences to the distinct mechanisms of action of the two antibiotics[39]. Under ampicillin selection, receiver cells experience disruptions in cell wall synthesis, which can lead to cell death after several replication cycles. Phagemid infection rescues these cells by providing antibiotic resistance, allowing them to survive despite ampicillin's effects. In contrast, gentamicin inhibits protein synthesis, making it more difficult for phagemid infection to rescue receiver cells, as the expression of the antibiotic resistance gene depends on protein synthesis.

As an application of this NOT gate circuit, we built a genetic circuit capable of detecting the presence of the sender cells in a co-culture. This circuit is based on two independent NOT gates (pHK001.23), where sgRNA-2 represses sfGFP and sgRNA-3 represses mCherry allowing the detection of two distinct sender cell populations independently (Fig. 2G). We tested the sensitivity of this circuit by co-culturing receiver cells at $OD_{600}$ 0.5 ($3.2 \pm 0.375 \times 10^5$ cfu) with sender cells sending sgRNA-2 at different densities (Fig. 2G). We found that this circuit could reliably detect sender cells at concentrations approximately 1000 times lower than receiver cells (approximately 300 sender cells among 300,000 receiver cells) within 4 h of co-

culturing. Notably, this high sensitivity remained unaffected even in the presence of a second sender cell population expressing a different sgRNA (sgRNA-3) at $OD_{600}$ 0.5.

## Inducible phagemid transfer

In Fig. 2, the input for the inverter/NOT gate was the presence or absence of sender cells. However, to build more complex biocomputing circuits, we need sender cells that can initiate the phagemid transduction upon the addition of an inducer signal. To achieve this, we designed inducible phagemid production in the sender cells by regulating the expression of protein VIII (P8) of the M13 phage gene cluster. This protein constitutes the main component of M13 phage, is highly expressed, and the phage progeny production depends on its expression[33,40,41]. We started by deleting gene VIII (gVIII) from the helper plasmid HP17_KO7, thus creating a new helper plasmid HP17ΔP8. Indeed, when we co-cultured sender cells with gene VIII deleted, the message phagemid was no longer transduced to the receiver cells (Fig. S2A). Next, we replaced gene VIII in HP17_KO7 with sfGFP. This gave us an approximation of the expression level of P8 in our system (Fig. S2B). We compared this expression level with sfGFP under the regulation of several inducible promoters ($P_{tet}$, $P_{tac}$, $P_{lux}$, and $P_{bad}$) on a pCDF plasmid (Fig. S2C). All four promoters tested were able to reach the expression level of sfGFP at the gene VIII locus upon induction. However, they differed in their leakiness in the absence of inducer, with $P_{lux}$ and $P_{tac}$, showing the highest leakiness. It is worth noting that the sfGFP expression under the control of $P_{bad}$ was delayed for approximately 50 min compared to the other promoters.

We then created a set of second helper plasmids (pHK316 – pHK346, Table S2) using a pCDF backbone where we placed the gene VIII downstream of an inducible promoter ($P_{tet}$, $P_{tac}$, $P_{lux}$, or $P_{bad}$). Thus, they conditionally express P8 in the presence of a chemical inducer, and the phagemid transfer should only occur when the sender cell is chemically induced. Next, we transformed sender cells with the first helper plasmid (HP17ΔP8), a second helper plasmid (pHK316 – pHK346) and a message phagemid (pBR322-based) (Fig. 3A). Upon co-culturing the inducible sender cells with the NOT gate receiver cells at a 3:1 ratio, sfGFP was repressed in the presence of the appropriate inducer (Fig. 3B). Initially, the $P_{bad}$-P8 construct (pHK346) did not show sufficient repression of sfGFP upon induction with 1% arabinose (Fig. 3B). This observation together with the previous observation of delayed sfGFP expression under the $P_{bad}$ promoter led us to hypothesize that the presence of the arabinose metabolic genes *araBAD*[42] is a problem. Thus, we deleted the *araBAD* genes from the chromosome of the JM101 strain. Using the JM101 Δ*araBAD* strain as the sender cell, we were also able to regulate phagemid transfer with arabinose (Fig. 3B). We also confirmed that the orthogonality is maintained in the

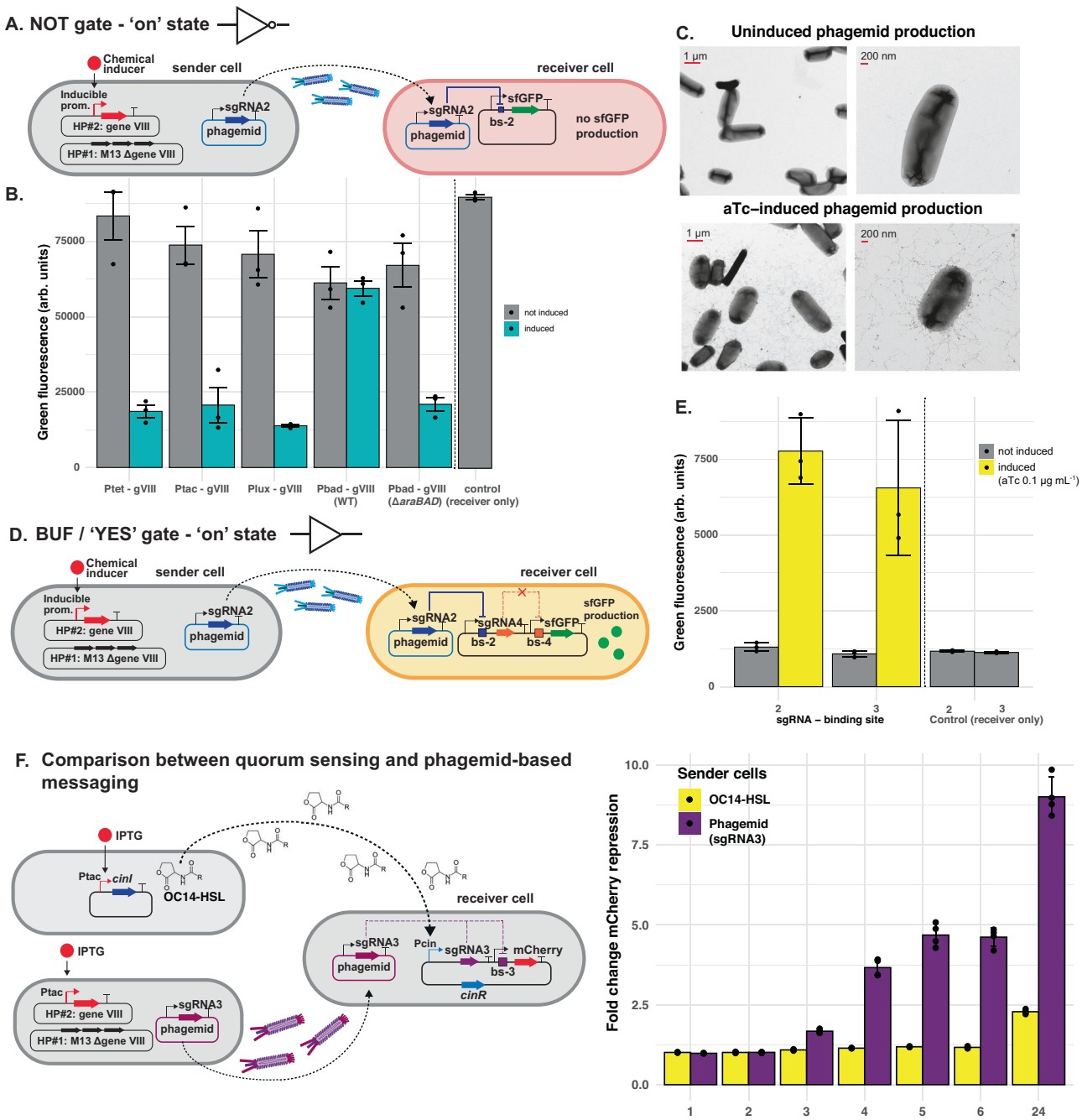

**Fig. 3 | Establishing inducible phagemid transfer in one-input logic gates.**
**A** Schematic representation of sender and receiver cells for an inducible NOT gate. Gene VIII (gVIII) was deleted on helper plasmid (HP) #1 and added to HP#2 under the control of an inducible promoter **B** Performance of different inducible promoters regulating P8 production. Green fluorescence (arbitrary units, arb. units) of the receiver cells was measured after 4 h of co-culturing with inducible sender cells with the indicated promoters controlling expression of gVIII in the presence (green) or absence (grey) of the corresponding inducer: aTc (0.1 mg liter$^{-1}$), IPTG (1 mM), AHL (10 mM) and arabinose (1%). The initial sender-to-receiver ratio was 3:1. Data represent the mean ± SD of 3 independent biological replicates. WT wild-type (strain JM101), ΔaraBAD: strain JM101 with genes araBAD deleted. **C** TEM images of sender cells in the presence or absence of 1 mg liter$^{-1}$aTc inducer, grown for 4 h at 37 °C with 1000 rpm shaking. **D** Schematic representation of one version of an inducible BUF/'Yes' gate. **E** Performance of two BUF gate variants using inducible phagemid sender cells. Green fluorescence (a. u.) of the receiver cells was measured

after 4 h of co-culturing with inducible sender cells (aTc-inducible) for BUF gates with sgRNA-2 transduced (as indicated in **D**). A second variant was tested in which sgRNA-3 is transduced and sgRNA-5 represses sfGFP. The initial sender-to-receiver ratio was 3:1. Data represent the mean ± SD of 3 independent biological replicates. **F** Comparison of phagemid-based to quorum-sensing-based intercellular communication in liquid culture. Receiver cells contained a mCherry reporter that can be repressed by sgRNA-3. This sgRNA-3 is produced if the receiver cells sense OC14-HSL or if they receive a message phagemid containing sgRNA-3. The inducible phagemid sender cells contained P$_{tac}$-regulated gVIII, whereas the quorum-sensing sender cells contained P$_{tac}$-regulated cinI for OC14-HSL production. Red fluorescence (arb. units) of the receiver cells was monitored at 1, 2, 3, 4, 5, 6, and 24 h of co-culturing (sender-to-receiver ratio 3:1) with inducible sender cells (IPTG-inducible phagemid transfer or OC14-HSL production). mCherry-fold repressions were calculated from these measurements. Data represent the mean ± SD of 4 independent biological replicates. Source data are provided as a Source Data file.

inducible phagemid transfer, with different sgRNA-binding site pairs giving 9–25-fold repression (Fig. S2E). To confirm visually that the M13 phage production is inducible, we imaged the sender cells in the presence and absence of inducer (anhydrotetracycline, aTc) with transmission electron microscopy (TEM). Indeed, we observed a dense lawn of phage particles in the induced state, and only a few scattered M13 phages in absence of the inducer (Fig. 3C).

Using the inducible phagemid transfer, we next engineered a BUF gate (or 'Yes' gate). We did this by adding another NOT gate on the receiver plasmid. Specifically, we placed the sfGFP reporter gene downstream of a promoter J23100 and a binding site for sgRNA-4. SgRNA-4 is present on the receiver plasmid and its production can in turn be repressed by sgRNA-2 encoded on the message phagemid. While in this BUF gate sfGFP expression is controlled by sgRNA-4 and sgRNA-2, we also built a second version with sgRNA-5 and sgRNA-3. For the two versions, we measured 4.7-5.8-fold activation (Fig. 3E), while orthogonality was maintained (Fig. S2F).

Next, we sought to compare the performance of our phagemid-based intercellular communication to the well-established quorum-sensing-based communication in liquid culture. To do this, we designed a new circuit (pHK001.3QS), which is based on a NOT gate where repression is mediated by sgRNA-3 (Fig. 3F). This circuit contains a sgRNA-3 under the regulation of the CinR/Pcin system, which is OC14-HSL inducible. We selected this quorum-sensing system for its good dynamic range and tight control[43,44]. The repression of mCherry can thus be triggered by the presence of a phagemid encoding sgRNA-3 or by the presence of OC14-HSL in the medium. As shown in Fig. S3G, the presence of sender cells constitutively producing phagemids resulted in a comparable repression of mCherry to that achieved by adding 10 µM OC14-HSL to the medium.

After confirming that the circuit functioned correctly for both modes of communication, we compared their efficiency using two different inducible sender cells: one in which M13 phagemid transfer was inducible via IPTG ($P_{tac}$-gVIII/pHK326) and another in which IPTG induced *cinI* expression (pHK-*cinI*), leading to OC14-HSL production. We monitored the repression of mCherry over time following induction at the start of co-culturing. While phagemid transfer led to mCherry repression within 3-4 h of co-culturing, quorum-sensing-based cell-cell communication required overnight co-culturing to achieve just a twofold repression, which is much lower than the response observed with the phagemid-based communication system (Fig. 3F).

## Two-input Boolean logic gates

Having established the intercellular communication system for single-input gates, we extended this approach to construct two-input Boolean logic gates, meaning that we had two sender cell populations transmitting phagemids to the receiver cells. We began with a NOR gate (Fig. 4A). In our design, the receiver plasmid contains two distinct sgRNA-binding sites (bs-2 and bs-3) positioned downstream of a promoter (J23100) that controls sfGFP reporter expression. sgRNA-2 and sgRNA-3 are placed on the two message phagemids. Consequently, sfGFP is expressed in the absence of inducer, but repressed in the presence of one or both inducers. Indeed, when we used aTc and isopropyl β-D-1-thiogalactopyranoside (IPTG)-inducible phagemid transfers of message phagemids (coding for sgRNA-2 and sgRNA-3, respectively) from the sender to the receiver cells, we observed sfGFP fluorescence only in absence of IPTG and aTc. When receiver cells were exposed to sender cells delivering phagemids encoding sgRNA-1 or sgRNA-6, no repression was observed, confirming the orthogonality of the NOR gate (Fig. S3A). To compare and report the quality (Q-score or circuit quality score) of logic gates, we assessed them by dividing the lowest 'ON' state by the highest 'OFF' state[45]. Using this approach, our NOR gate has a Q-score of 175.4.

Next, we constructed an OR gate. To achieve this, we took the NOR gate but added a signal inverter (NOT gate) to the receiver plasmid (Fig. 4B). In this circuit, sgRNA-2 and sgRNA-3 do not control sfGFP directly, but they control the expression of sgRNA-4, that in turn controls repression of sfGFP. Induction with aTc, IPTG, or both increased sfGFP expression by 13.7-fold, 10.2-fold, and 22.6-fold, respectively, resulting in a Q-score of 10.2 (Fig. 4B). Again, orthogonality was maintained when receiver cells were co-cultured with sender cells constitutively delivering sgRNA-1 or sgRNA-6 (Fig. S3B).

Our next goal was to construct an AND gate. Compared to the NOR gate's receiver plasmid, we inverted the signal of the incoming phagemids. More specifically, in this circuit, the inducible transduction of sgRNA-2 represses sgRNA-4, and transduction of sgRNA-3 represses sgRNA-5 (Fig. 4C). Both sgRNA-4 and sgRNA-5 repress the sfGFP reporter gene in the receiver cells. Therefore, sfGFP is produced only if both sgRNA-2 and sgRNA-3 are transduced. The proper functioning of this AND gate required some fine-tuning. In particular, we optimized the promoter strengths on the message phagemids and the inducer concentrations (Fig. S3C, D). Under the optimized conditions (0.025 mg liter$^{-1}$ aTc and 0.25 mM IPTG, with J23119 promoter controlling message sgRNAs production), we observed an 11.5-fold induction of sfGFP expression for the condition where an output is expected, that is in presence of both inducers (Fig. 4C). We also detected a 4.3-fold increase in sfGFP expression with aTc alone, and thus resulting in a lower Q-score of 2.6. We attribute this to the leakiness of the $P_{tac}$ promoter, which we had noticed previously (Fig. S2D).

Next, we wanted to build a NAND gate. This required inducing P8 repression (instead of P8 activation) in the sender cells. We achieved this by adding the transcription factor cI as inverter to the helper plasmid (Fig. S3E, F). cI is regulated by inducible promoters ($P_{tac}$ or $P_{tet}$). In the presence of inducers (IPTG or aTc), cI is expressed and it represses expression of P8. We first individually tested both versions of these sender cells (containing $P_{tac}$ or $P_{tet}$) sending phagemids containing either sgRNA-2 or sgRNA-3 with NOT gate receiver cells (Fig. S3E, F). As expected, we observed high sfGFP expression in the presence of inducers, but repression in their absence. To create a two-input NAND gate, we combined the two inverter sender cells with the receiver cells already used in the OR gate (Fig. 4D). Testing this circuit revealed that sfGFP was indeed expressed in the presence of no or one input and only repressed (approximately 10.4-fold) when both IPTG and aTc were added to the culture, resulting in a Q-score of 7.3 (Fig. 4D).

## Four-input logic gates

To test the scalability and robustness of our approach, we extended it to build four-input logic gates. To achieve this, we created two hybrid inducible promoters by adding an extra operator site downstream of the promoters, namely tetO downstream of $P_{lux}$ and lacO downstream of $P_{bad}$, as well as the corresponding regulatory proteins into the helper plasmid. These hybrid inducible promoters require two inducers for activation: acyl homoserine lactone (AHL) and aTc for $P_{lux}$-tetO, and arabinose and IPTG for $P_{bad}$-lacO. We then placed gene VIII downstream of these promoters ($P_{lux}$-tetO_gene VIII and $P_{bad}$-lacO_gene VIII), making its expression dependent on the presence of both inducers (Fig. 5A). In addition, we also engineered inverter sender cells using hybrid promoters to regulate cI protein expression ($P_{lux}$-tetO_cI and $P_{bad}$-lacO_cI), which in turn represses the P8, thereby blocking phagemid transfer in the presence of both inducers (Fig. S4C, D). Again, we first individually tested all versions of these sender cells (containing $P_{lux}$-tetO_gene VIII and $P_{bad}$-lacO_gene VIII) sending phagemids containing either sgRNA-2 or sgRNA-3 with NOT gate receiver cells. Indeed, sender cells containing helper plasmids with $P_{lux}$-tetO_gene VIII or $P_{bad}$-lacO_gene VIII successfully delivered their phagemids

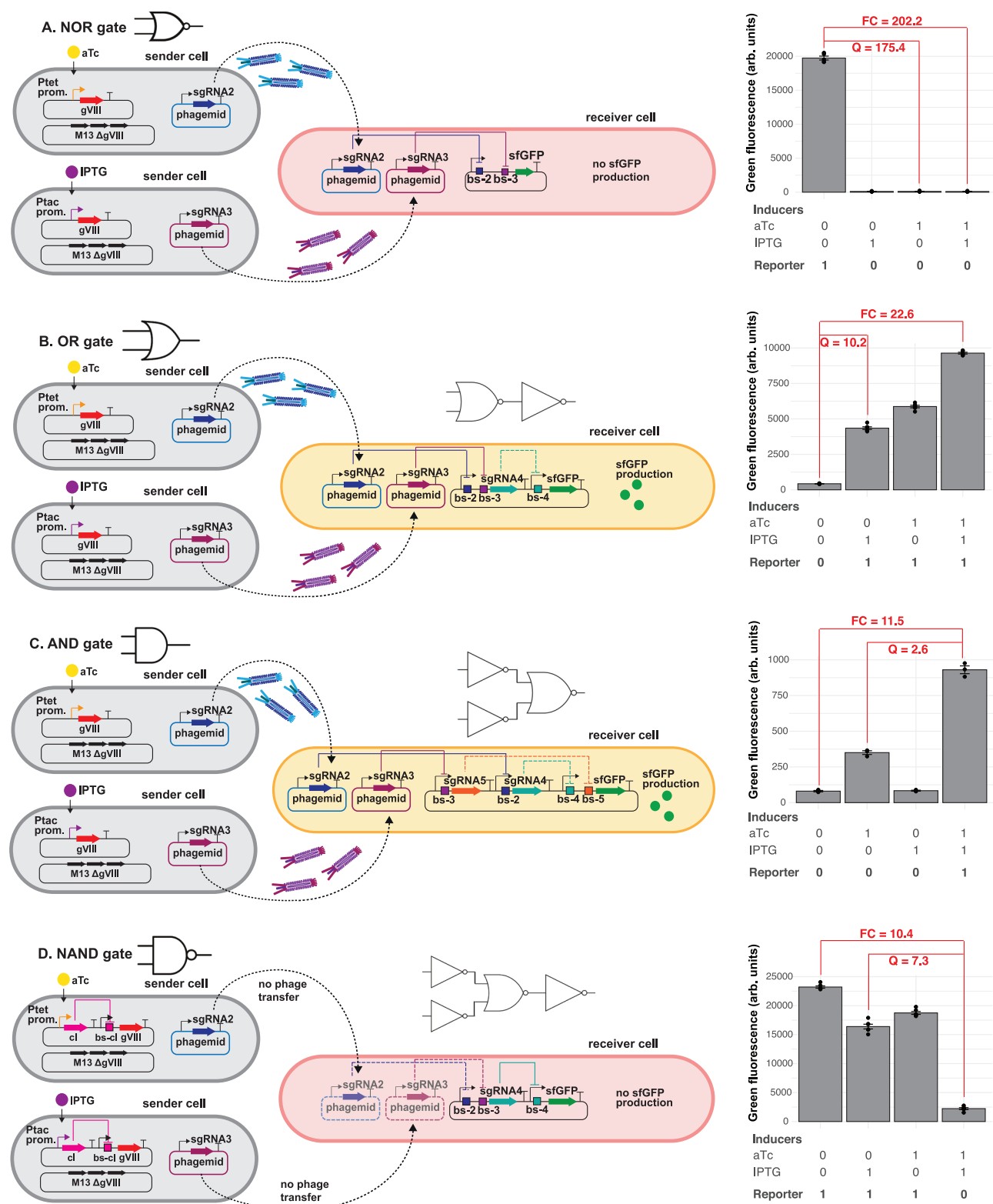

only when both inducers were present (Fig. S4A, B), while for the constructs with the cI inverter, phagemid transfer was inhibited when both inducers were present (Fig. S4C, D).

After confirming the functionality of the hybrid inducible promoters, we constructed a four-input AND gate by layering two sender cells, each representing a two-input AND gate to control the transfer of phagemids containing sgRNA-2 or sgRNA-3 with the receiver cell previously used for the two-input AND gate. When all four inputs were present [1111], sfGFP expression increased 9-fold, consistent with an AND gate's truth table, while other combinations did not exceed a 2.6-fold increase, resulting in a Q-score of 3.4 (Fig. 5A). Similarly, using these hybrid promoters to repress gene VIII (indirectly via cI), we constructed a four-input NAND gate by layering two sender cells (each a two-input NAND gate) with the receiver cells from the OR gate. With all four inputs [1111], sfGFP expression was repressed 17.5-fold, aligning with the NAND gate's truth table, while other inducer combinations yielded a maximum of 1.2-fold repression, resulting in a Q-score of 14.9 (Fig. 5B).

**Fig. 4 | Two-input Boolean logic gates in multicellular consortia. A** 2-input NOR gate. Left: **A** schematic representation of an inducible NOR gate design. Right: Average of green fluorescence (arb. units) measured by flow cytometry in the presence or absence of 0.1 mg liter⁻¹ aTc and 1 mM IPTG inducers. Data represents the mean ± SD of 5 independent biological replicates. **B** 2-input OR gate. Left: A schematic representation of an inducible 'OR' gate design. Right: Average of green fluorescence (arb. units) measured by flow cytometry in the presence or absence of 0.1 mg liter⁻¹ aTc and 1 mM IPTG inducers. Data represents the mean +/- SD of 6 independent biological replicates. **C** 2-input AND gate. Left: A schematic representation of an inducible 'AND' gate design. Right: Average of green fluorescence (arb. units) measured by flow cytometry in the presence or absence of 0.025 mg

liter⁻¹ aTc and 0.25 mM IPTG inducers. Data represents the mean ± SD of 3 independent biological replicates and error bars represents standard deviation. **D** 2-input NAND gate. Left: A schematic representation of an inducible 'NAND' gate design. Right: Average of green fluorescence (arb. units) measured by flow cytometry in the presence or absence of 0.1 mg liter⁻¹ aTc and 1 mM IPTG inducers. Data represents the mean ± SD of 6 independent biological replicates. For all 2-input gates, the initial sender-to-receiver ratio was 3:1. Fold-change (FC) is calculated by dividing the 'ON' state by the 'OFF' state, while the quality score (Q) was calculated by dividing the lowest 'ON' state by the highest 'OFF' state. Source data are provided as a Source Data file.

## Discussion

Synthetic biology and its applications are poised to drive the next industrial revolution[1–4]. It is becoming increasingly evident that distributing the circuitry across different strains within engineered consortia will further enhance the complexity of information processing achievable with synthetic circuits[14,24,46,47]. To realize this potential, establishing orthogonal and reusable communication channels for multi-layered and simultaneous intercellular communication is crucial[24,27,28].

Here, we expanded the toolkit for orchestrating complex information processing in synthetic microbial consortia. We successfully integrated CRISPRi with phagemid transfer to establish an intercellular communication system. Building on the work of Ortiz and Endy[31], we first tested the transfer of sgRNAs by consititutively produced phagemids. We also leveraged the phagemid transfer system to detect the presence of highly diluted sender cells in a co-culture (Fig. 2G). Importantly, our system not only enables repression and activation in receiver cells but also allows conditional regulation of phagemid message transfer (Fig. 3). This system allows us to precisely control communication, using chemical inducers to either initiate (Fig. 3) or block phagemid transfer (Fig. 4D). We achieved high transfer rates (>97% for both constitutive and inducible phagemid transfer) without any selection applied. Simultaneously, another team independently explored phagemid-mediated intercellular CRISPRi for biocomputation in bacterial consortia[48]. Their findings were consistent with ours, highlighting the robustness and potential of this approach. Their detailed characterization of secretion and infection rates provides valuable complementary insights. However, a key distinction of our approach lies in our ability to control phagemid transfer, enabling the targeted distribution of computational tasks, such as constructing 2-input NAND gates (Fig. 4D) and complex 4-input gates (Fig. 5).

We also demonstrated that in liquid culture, phagemid-based communication led to a faster and stronger response than quorum-sensing-based communication in a comparable setup (Fig. 3F). We attribute this difference between the two systems to the dilution of the secreted quorum-sensing molecules in liquid culture, whereas a single successfully transferred phagemid is sufficient to produce a robust response in the receiver cell. This aligns with the high sensitivity of phagemid detection demonstrated in Fig. 2G. While quorum-sensing-based intercellular communication systems have been shown to work effectively between close-by colonies on agar plates[27,49] or on conditioned media for receiver activation[50,51], our results highlight that phagemid-based communication constitutes a highly efficient intercellular signalling mechanism in liquid culture.

Our approach is highly modular: the same message phagemids were re-used for all gates and the difference between the two- and four-input AND and NAND gates was only the inducible promoters. Thus, different and more complex gates can easily be built by combining different sender and receiver cells with no to minimal changes and fine-tuning. Although we have not yet implemented the full set of logic gates using our phagemid-based intercellular communication, we have designed two more two-input gates, namely NOR and XNOR

gates (Fig. S13). In these designs, the sender cells closely resemble those used in the four-input NAND gate, while the receiver cells are identical to those in the NOR and OR gates, respectively.

Taking advantage of highly orthogonal sgRNAs for CRISPRi, we observed minimal crosstalk in our logic gates. As a comparison, a four-input AND gate that was constructed in a single bacterial cell used three circuits that integrated four inducible systems, utilizing 11 regulatory proteins[7]. Moreover, directed evolution had to be applied to increase the dynamic range and orthogonality of the circuits[7]. Unlike for intercellular communication based on quorum-sensing molecules; by establishing inducible phagemid transfer, we can effectively separate the inputs (chemical inducers and its corresponding regulatory proteins) from the "wiring" (phagemid transfer). In addition, we were able to carry out all our experiments in liquid culture, where some other multicellular computing approaches rely on arranging the individual consortium members in a specific spatial pattern[27,49]. Finally, DNA-based intercellular communication also allows for easier diversification of messages, simply by modifying the sgRNA sequence. It should be straightforward to build our designs with different sgRNA sequences.

However, our current setup still has some limitations and offers room for further improvement. One of them is asymmetric growth rates of sender and receiver cells. Phagemid regulation and production imposes a significant burden on sender cells. To ensure complete phagemid transfer to the receiver cell population, we increased the sender-to-receiver cell ratio from 1:1 or 2:1 to 3:1. Future work will include the integration of the helper plasmid (HP17ΔP8) into the chromosome, thereby reducing the load and burden.

Another challenge is dCas9 competition in the receiver cells[52]. The different sgRNAs in the receiver cells all bind to the finite pool of dCas9. We found that using a medium copy number for the sender phagemids (15–60 copies per cell) ensured an adequate transfer rate without overwhelming the dCas9 protein in the receiver cell. However, for future more complex circuits where even more different sgRNAs might end up in the receiver cells, dCas9 competition might become an issue. While increasing the expression of the currently used dCas9 is limited by its toxicity[53], less toxic dCas9 variants have been described[54] and might allow higher expression, thus reducing the competition effect. Another future avenue is the integration of CRISPR activation (CRISPRa)[22,55] into the gene regulatory circuits. Using direct activation instead of two inverters would simplify the circuits in the receiver cells for certain computations (e.g., for the OR gate).

In our designs, we have exclusively used CRISPRi-based gene regulation in receiver cells while leveraging well-established transcription factors to control inducible promoters and invert signals in sender cells. This approach allowed us to avoid expressing dCas9 in sender cells while fully utilizing dCas9 in receiver cells, where it was already required to process the transferred message. However, we believe that our framework of phagemid-delivered sgRNA messages is highly flexible and can integrate both CRISPR-based gene regulation and traditional transcription factor logic—such as that used in the Cello framework[8] —in both sender and receiver cells. Additionally, carrying out part of the computation in receiver cells with

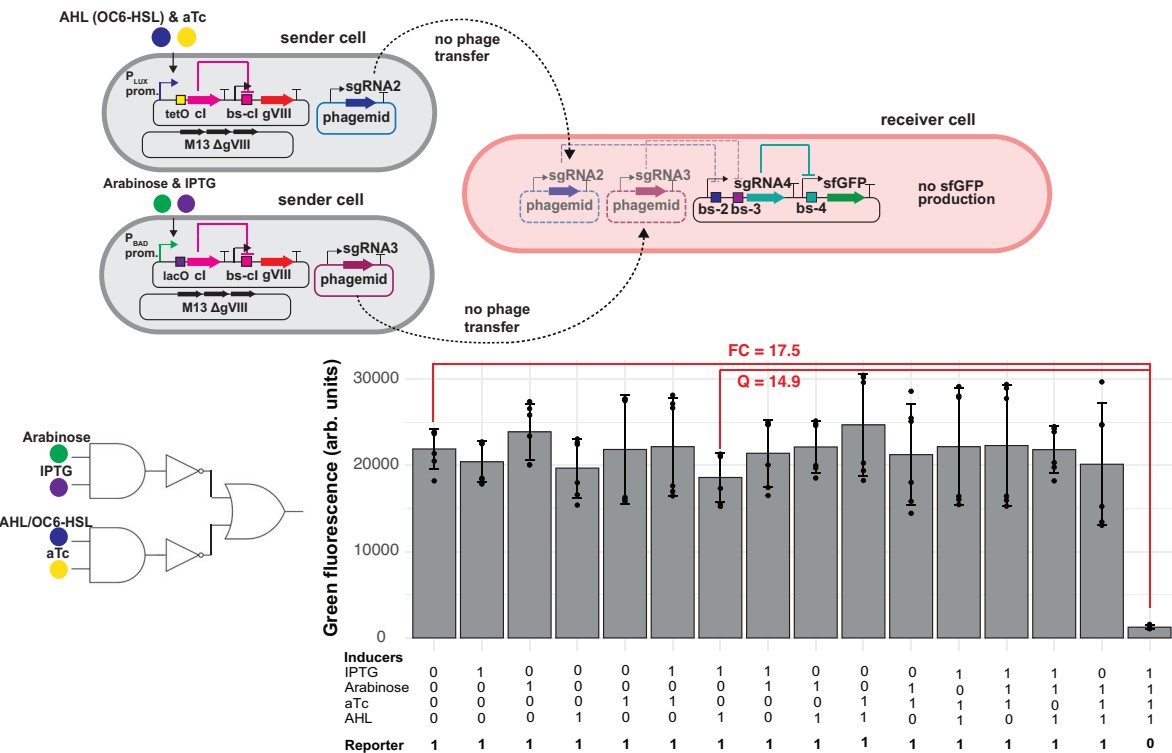

**Fig. 5 | Four-input Boolean logic gates design. A** 4-input AND gate. Top: 4-input 'AND' gate design: a combination of two-input 'AND' gates in sender cells and a two-input 'AND' gate in receiver cells. Bottom: Mean of green fluorescence (arb. units) ± SD of 6 independent replicates, measured by flow cytometry in the presence or absence of 1 mM IPTG, 1 % arabinose, 0.1 mg liter⁻¹ aTc and 1 mM AHL inducers. **B** 4-input NAND gate. Top: 4-input 'AND' gate design: a combination of two-input 'AND' gates in sender cells, two inverters and a two-input 'OR' gate in receiver cells.

Bottom: Mean of green fluorescence (arb. units) ± SD of 6 independent replicates, measured by flow cytometry in the presence or absence of 1 mM IPTG, 1 % arabinose, 0.1 mg liter⁻¹ aTc and 1 mM AHL inducers. For both 4-input gates, the initial sender to receiver ratio was 3:1. Fold-change (FC) is calculated by dividing the 'ON' state by the 'OFF' state, while the quality score (Q) was calculated by dividing the lowest 'ON' state by the highest 'OFF' state. Source data are provided as a Source Data file.

transcription factors could help mitigate the issue of dCas9 competition mentioned earlier.

In the circuits presented in this paper, all messages end up in the same receiver cell. This allowed us to establish and optimize the combination of M13 phagemid-based intercellular communication with gene regulation using CRISPRi. Furthermore, we demonstrated that this communication system is more sensitive than a comparable quorum-sensing-based cell-cell communication in liquid culture (Fig. 3F). It allowed us to build four-input AND and NAND gates (Fig. 5) matching the complexity of logic gates built so far in monoculture[7]. However, in future we will take full advantage of multicellular computing and also distribute the output signal across different cell types[24]. This will make it possible to carry out complex computation such as addition or multiplication of numbers beyond what has been achieved in monoculture[10]. Given the availability of sgRNA design tools[56,57] and the successful use of up to 24 orthogonal and multiplexed sgRNAs[58], we anticipate that our method holds significant potential for scaling up to more complex biocomputing systems. Finally, combining phagemid-based cell-cell communication with other communication channels, such as chemical-based signals (including diffusible chemicals[27] and volatile compounds[59]) and other DNA-based communication methods (such as conjugation[28–30]), presents another intriguing avenue to explore.

In conclusion, our results demonstrate that phagemid-mediated intercellular communication combined with CRISPR-based gene regulation is a promising strategy for carrying out multicellular computing. We hope our work paves the way for engineering microbial consortia that carry out complex computations for applications in fields as diverse as biocomputing, bioproduction, bioremediation, biosensing, diagnostics and therapeutics[24,47,60].

## Methods

### Strains and culture conditions

The strains used in this paper are listed in Table S1. The helper phagemid HP17_KO7 was a gift from Hendrik Dietz (Addgene plasmid #120346)[37]. All co-culture experiments for phagemid transfer were conducted using *E. coli* strain JM101 (glnV44 thi-1 Δ(lac-proAB) F'[lacIqZΔM15 traD36 proAB+])[35,36] or JM101 Δ*araBAD* (Table S1). Plasmid and phagemid cloning were performed using *E. coli* strain DH5α. *E. coli* strains were cultivated in LB medium at 37 °C with shaking at 200 rpm, unless stated differently. For solid cultivation, 1.5% (wt/vol) agar was added to LB medium. When required, gentamicin (25 mg liter⁻¹) (Sigma-Aldrich cat. no. G1914), ampicillin (100 mg liter⁻¹) (Sigma-Aldrich cat. no. A9518), kanamycin (50 mg liter⁻¹) (Sigma-Aldrich cat. no. K4000), spectinomycin (50 mg liter⁻¹) (Sigma-Aldrich cat. no. S4014), arabinose (1% w/v) (Sigma-Aldrich cat. no. A3256), anhydrotetracycline (aTc) (0.1 mg liter⁻¹) (Chemodex cat. no. 13803-65-1), N-(β-Ketocaproyl)-L-homoserine lactone (AHL/OC6-HSL) (10 μM) (Sigma-Aldrich cat. no. K3007), N-(3-Oxotetradecanoyl)-L-homoserine lactone (AHL/OC14-HSL) (10 μM) (Sigma-Aldrich cat. no. O9264), and Isopropyl β-D-1-thiogalactopyranoside (IPTG) (1 mM) (Sigma-Aldrich cat. no. PHG0010) were added to the media.

### PCR and cloning methods

PCRs were performed using Phusion polymerase (Thermo Fisher) according to the manufacturer's manual. Oligonucleotides used in this paper (Table S3) were purchased from Sigma-Aldrich. PCR products were analyzed by gel electrophoresis on 1% (wt/vol) Tris-borate-EDTA (TBE) agarose containing 1x SYBR Safe (Thermo Fisher) at 135 V with 1x TBE running buffer for 15 min. Deletion of *araBAD* genes from the chromosome of strain JM101 was performed using the lambda red recombination system[61].

The plasmids and phagemids used in this study are listed in Table S2 and the parts used to construct the circuits are listed in Table S4. Plasmid maps are available in Figs. S6–S10. We constructed and cloned plasmids and phagemids using our previously described assembly method for synthetic genetic circuits[62]. Briefly, the cloning method consists of two main steps: Step 1 involves the Gibson assembly of transcriptional units into individual intermediate plasmids. All DNA parts are designed with the same Prefix (CAGCCTGCGGTCCGG) and Suffix (TCGCTGGGACGCCCG) sequences to facilitate modular Gibson assembly. Forward and reverse primers that anneal to these Prefix and Suffix sequences are used in PCRs to add unique linkers to the DNA parts. The PCR amplifications are then column-purified using the Monarch PCR & DNA Cleanup Kit (NEB) and assembled using NEBuilder HiFi DNA Assembly Master Mix (NEB) at 50 °C for 1 h. These DNA parts are inserted into backbones that were previously digested with corresponding restriction enzymes (NEB) at 37 °C for 1 h, resulting in intermediate plasmids containing individual transcriptional units. Step 2 involves the digestion of these intermediate plasmids with restriction enzyme sets that create overlapping sequences. These plasmids are then purified and assembled as in Step 1 to form a single plasmid containing the final genetic circuit. Subsequently, 1 μl of the non-purified Gibson assembly reaction is transformed into 50 μl of electrocompetent DH5α cells. Finally, these transformed cells are plated onto selective agar plates.

### Phagemid transfer assay

Sender cells were JM101 strains with or without deletion of *araBAD* genes that contained M13 phage helper plasmid(s) and a message phagemid (Table S1). Receiver cells were JM101 strains that contained the reporter plasmid and pJ1996v2[21] plasmid encoding dCas9 and csy4 nuclease. Schematic representation of the phagemid transfer assay can be found in Fig. S5. We inoculated both sender and receiver cells from single colonies (at least three biological replicates for each sample) and grew them overnight at 37 °C with 1000 rpm shaking in 400 μL of 2x Yeast Extract Tryptone medium (2x YT) with appropriate antibiotics added. We diluted these overnight cultures ten-fold for receiver cells and five-fold for sender cells into fresh 2x YT media (1 mL). We grew the cultures at 37 °C with 1000 rpm shaking for 1 h (early log phase). Afterwards, we measured ODs of the receiver cell cultures and adjusted them to an OD600-0.5. For sender cells, we pelleted the cells with centrifugation and removed the supernatant. For sender cells, we pelleted the cells with centrifugation and removed the supernatant. We resuspended the cell pellets with 2× YT containing kanamycin 50 mg liter⁻¹ (for constitutive phagemid transfer) or 2× YT containing kanamycin 50 mg liter⁻¹ and spectinomycin 50 mg liter⁻¹ (for inducible phagemid transfer) and adjusted samples to an OD600-0.5, thus removing already produced phagemids and ensuring that the experiment will be carried without selection pressure for message transmission. We mixed sender and receiver cells at 1:1 or 2:1 ratio for experiments with constitutive phagemid production and at a ratio of 3:1 for inducible phagemid production. We added 40 μL of this mixture, chemical inducer(s) (as indicated), and 360 μL of 2× YT with kanamycin 50 mg liter⁻¹ (for constitutive phagemid transfer) or 2× YT with kanamycin 50 mg liter⁻¹ and spectinomycin 50 mg liter⁻¹ (for inducible phagemid transfer) into a 2 mL 96x deep-well plate. The 96× deep-well plate was then covered by Breathe-Easier sealing membrane (Merck, cat. no. Z763624). We incubated the deep-well plate at 37 °C with 1000 rpm shaking for 4–6 h. Following the incubation, we diluted the samples 200 times with 1× PBS (pH 7.4) and analyzed them using a Novocyte Flow cytometer.

### Flow cytometry and data analysis

We measured the fluorescence of the samples from phagemid transfer assay using a Novocyte Flow cytometer. We used 488 nm excitation laser in combination with FITC filter (emission 530 nm) for sfGFP measurements and 561 nm excitation laser in combination with PE Texas Red filter (emission 615 nm) for mCherry measurements. Flow cytometry raw data were recorded using NovoExpress software

(version 1.6.2). First, to discriminate between cells and other particles, all measured events were gated by forward scatter height (FSC.H) > 1000 arbitrary unit (a.u.) and side scatter height (SSC.H) > 200 a.u. Second, we excluded doublets by plotting the FSC.H against the forward scatter area (FSC.A) and set a gate for the events with approximately 1:1 ratio of FSC.H to FSC.A. We recorded 30,000 events of singlet cells. We then set a gate to select for the receiver cells ('sender/receiver threshold' in Fig. 2C) for red fluorescence (PE.Texas.Red.H) above 2000 a.u. Finally, the data were analysed and visualized in R using RStudio 1.4.1106 (R 3.4.0). The reported GFP fluorescence values were calculated as the geometric mean of individual biological replicates. Then we subtracted the basal green fluorescence value of the cells, which was measured from the non-fluorescence JM101 control. GFP reporter fold change (FC, Figs. 4 and 5) was then determined by dividing the mean GFP fluorescence levels of the intended 'ON' state by those of the 'OFF' state. For calculating the fraction of receiver cell population that received the phagemid messages (Fig. 2D), we set additional gates for the 'ON' state and 'OFF' states by setting a threshold for green fluorescence (FITC.H) above or below 10,000 a.u., respectively (Fig. S12).

### Microplate reader experiments

To determine the necessary expression level of P8, we measured fluorescent reporter gene expression by measuring fluorescence with a microplate reader (Fig. S2B, C). In this experiment, 2 ml of selective LB was inoculated with single colonies in a falcon tube and incubated at 37 °C for approximately 6 h with 200 rpm shaking. The cells were then pelleted at 4200 rcf and resuspended in selective EZ medium (Teknova) containing 0.4% glycerol. We added 120 µl of 0.05 $OD_{600}$ bacterial suspensions per well in a 96-well CytoOne plate (Starlab) and supplemented it with inducers to reach the desired concentrations. The plate was covered with the supplied lid and were incubated at 37 °C with double-orbital shaking in a Synergy H1 microplate reader (Biotek) running Gen5 3.04 software. Fluorescence was measured after 16 h using the following settings for sfGFP: Ex. 479 nm, Em. 520 nm. Six biological replicates were measured for each sample. The fluorescence levels were processed as follows: (i) subtracting the fluorescence signal of a blank sample, (ii) dividing the resulting value by the absorbance at 600 nm to account for differences in bacterial concentration, and (iii) subtracting the bacterial autofluorescence of a strain without reporter genes. The normalized data were plotted using ggplot2 package in RStudio 1.4.1106 (running R 3.4.0).

### Transmission electron microscopy (TEM) assay

The sender cells containing the first helper plasmid (HP17ΔP8), the second helper plasmid (aTc-inducible P8), and a message plasmid (pBR322-backbone) were inoculated from single colonies and grown overnight at 37 °C with 1000 rpm shaking in 400 µl of 2x YT with ampicillin (100 mg liter$^{-1}$), kanamycin (50 mg liter$^{-1}$) and spectinomycin (50 mg liter$^{-1}$) added. We diluted this overnight cultures tenfold into two falcon tubes with fresh 2x YT media (1 mL), with and without aTc (0.1 mg liter$^{-1}$) added, and let the cultures grow at 37 °C with 1000 rpm shaking in 2x YT media for 4 h. Afterwards, the samples were washed with 1 ml of 1× PBS in an Eppendorf tube. After washing, the samples were pelleted by centrifugation at 7000 rcf, then gently resuspended in 1 ml of 1× PBS containing 2.5% formaldehyde to ensure homogeneity and fix the samples. The samples were incubated at 4 °C with gentle shaking on a rocker shaker (12 oscillations per minute) overnight to preserve the cell and phage structures. The following day, the fixed cultures were concentrated to 1/10th of their original volume by centrifugation at 4,000 rcf and removing the supernatant. The samples (20 µL) were placed on parafilm under TEM grids (carbon film-coated, 400 mesh, copper, Sigma-Aldrich) for 2 min to ensure direct contact. The grids were then washed three times with water and stained with 1% uranyl acetate for 30 s. Finally, the samples were examined under a Philips CM100 120 kV TEM microscope.

### Reporting summary

Further information on research design is available in the Nature Portfolio Reporting Summary linked to this article.

## Data availability

The source data underlying Figs. 2–5 are provided as a Source Data file. The plasmids used in this study (Table S2) and their annotated sequences are available through Addgene [https://www.addgene.org/Yolanda_Schaerli/] Addgene ID #235447-235486. Source data are provided with this paper.

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

## Acknowledgements

We thank Lucie Gilliéron, Camille Walti, and Marie Gaudard for excellent technical assistance and all Schaerli lab members for useful discussions. This work was funded by Swiss National Science Foundation Grant 310030_200532 (to Y.S.), a UNIL FBM PhD fellowship in Life Sciences (to R.A.) and the University of Lausanne.

## Author contributions

H.K. and Y.S. designed the research. H.K. performed experiments and analyzed data. F.Z. performed experiments. R.A. performed the transmission electron microscopy. Y.S. supervised the project. H.K. and Y.S.

wrote the manuscript. All authors approved the final version of the manuscript.

## Competing interests

The authors declare no competing interests.
