## [Peer Review file · Nature Communications]

Engineering Intercellular Communication using M13 Phagemid and CRISPR-based Gene Regulation for Multicellular Computing in *Escherichia coli*

Corresponding Author: Professor Yolanda Schaerli

Version 0:

Reviewer comments:

Reviewer #1

(Remarks to the Author)

The authors have produced an inducible M13 phage communication system, sending sgRNAs which repress transcription from engineered circuits in receiver cells. This was demonstrated by implementing digital logic bio-computation. The work is presented very clearly and is rather timely given preprints from other groups (highlighted here by the authors).

My only comments are minor details that need to be addressed.

- In figures 3B and 3E, can you provide a control to compare the receiver cell fluorescence without the sender cell present.
- Please report quality of each logic gate as lowest ON state divided by highest OFF state; this has become a useful convention for quickly understanding the efficacy of different bio-computation approaches.
- There are a few missing 2-input gates which I think should be included for completeness.
- Were the two 4-input gates the only ones that were attempted? Why were those particular gates chosen?
- Check references – 7 and 9 are the same
- Line 329 “optimzse” -> optimize
- Please include catalogue numbers for chemicals used in the methods – this is crucial for reproducibility.
- What did you use to cover the 96-well plates in the plate reader experiments?
- The fluorescence normalisation carried assumes in this manuscript assumes a linear relationship between autofluorescence and absorbance, which is not true in the GFP spectrum. See <https://pubs.acs.org/doi/full/10.1021/acssynbio.0c00296>

Reviewer #2

(Remarks to the Author)

In this work, Kusumawardhani, Zoppi, Avendaño, and Schaerli expand on Ortiz and Endy's earlier implementation of M13 phagemid-based intercellular communication [Ref. 1] by introducing two innovations: (1) allowing the inducible production of the phagemid by sender cells, and (2) coupling the phagemids to CRISPRi logic circuits in the receiver cells.

The manuscript is well written and the experimental work is well executed and communicated clearly. However, while it has the potential to do so, I feel that the current version of the manuscript does not provide the field with results of sufficient significance to justify publication in a broad-audience journal like Nature Communications. Below, I will describe my thoughts about how the authors might extend their current work to achieve this significance.

On the significance of the work:

My fundamental concern is that while the paper is focused on expanding the capabilities of DNA-based intercellular communication systems, most of the experiments presented in the paper do not engage in a meaningful way with either intercellular communication or distributed computation. The addition of chemical inducibility to Ortiz and Endy's M13 system is a legitimate advance that has utility for future efforts in engineering multicellular consortia, but this is not the case for the construction and characterization of CRISPRi logic circuits, which constitute the bulk of the work. This can be illustrated by

noting that all of the circuits in Figs 3-5 follow the same overall architecture:

Phagemid-based Circuit Architecture (this work):

1. The sender cells contain an inducible M13 channel and a constitutive sgRNA on the phagemid.
2. The receiver cell contains a latent CRISPRi circuit that takes the transferred sgRNAs as inputs.

The above architecture is logically identical to the following single-strain implementation:

Single-strain Circuit Architecture:

1. One cell strain contains the inducible sgRNA inputs, as well as the latent CRISPRi circuit that takes those sgRNAs as inputs.

There is no clear reason why one should choose to pick the multi-strain implementation over the single-strain implementation when implementing a circuit. Even if one insists on distributing the circuit between multiple cell types, there is no clearly presented benefit of using phagemids for intercellular communication as opposed to a logically equivalent implementation using conventional quorum sensing systems:

Quorum sensing-based Circuit Architecture:

1. The sender cells contains an inducible AHL synthase.
2. The receiver cell contains a latent CRISPRi circuit, as well as cassettes that inducibly express the sgRNA inputs for the CRISPRi circuit based on the presence of the cognate AHL.

Convincingly presenting the benefits of the phagemid-based multicellular implementation, over conventional multicellular implementations and/or over single-strain implementations, will be the central challenge of the authors in elevating their current work to a higher level of significance. This needs to be accomplished both through revisions on the narrative of the paper as well as the choice of experiments that are presented.

On comparing multi-strain implementations to single-strain implementations:

The ability to construct complex CRISPRi logic circuits in single-strain implementations is now well-established in the field [Ref. 2]. Transferring CRISPR components intercellularly to conduct gene editing or DNA cleavage in the recipient cells is also routinely done [Ref. 3], even in the context of CRISPRi [Ref. 4]. It is therefore unsurprising that a multicellular CRISPRi logic circuit, distributed in the way that is presented in the paper, would work.

The authors must therefore demonstrate that their system is able to create circuits that can achieve qualitatively different functionalities from conventional single-strain CRISPRi implementations.

For example, what if one were interested in building circuits that don't sense the concentrations of target inducer molecules, but rather the densities of target cell strains within a population? The receiver cell could then act as a dynamic reporter of the population composition of a multistrain consortium, with different logic gates used to give readouts of different population states. In such an implementation, each monitored cell could constitutively express phagemids containing a unique sgRNA input into the logic circuit. Building such a circuit would require additional characterization experiments that assess, e.g., the sensitivity of M13 detection (i.e., in a sender-receiver coculture, what is the threshold density of sender cells below which receivers can no longer reliably detect M13-driven communication?), but would constitute a type of circuit that takes advantage of the unique properties of multi-strain implementations.

On comparing phagemid-based communication to quorum sensing channels:

Ortiz and Endy highlighted the specific advantages of DNA-based communication over other forms of intercellular communication as (1) Message-channel decoupling, (2) Flexibility of encoded message, and (3) Specificity of message receipt. Marken and Murray [Ref. 5] also recently identified (4) Large information capacity of the message and (5) Dynamic mutability of the message as additional unique advantages of DNA-based communication. With the exception of message-channel decoupling, which is a structural property of the M13 phagemid system, none of the other unique advantages of DNA-based communication are leveraged in this work.

The authors do make the point that by transferring sgRNAs along phagemid channels, one can reach a total number of communication channels that is dictated by the (potentially very large) number of orthogonal sgRNA-binding site pairs, and empirically demonstrate 6 such pairs. However, the ability to reach up to 8 orthogonal intercellular channels has already been demonstrated without DNA-based communication [Ref. 6]. If the authors want to lean into the number of possible communication channels as a central advance of their work, they should empirically demonstrate the ability to scale far beyond the current non-DNA record of 8 orthogonal channels. There are many claims in the literature about the scalability of sgRNA-based systems, but if the authors were able to empirically demonstrate a dramatic increase in the number of simultaneous orthogonal communication channels this way, that would certainly constitute a major advance in the field.

An alternative approach is to conduct head-to-head characterizations of signaling performance between quorum sensing-based systems and the phagemid system, which to my knowledge has not been done. Characterizing communication-relevant properties such as the sensitivity of the receiver cells to the sender cell density (as I described above), the timescale

from the onset of signal transmission to receiver response, etc., and how they vary between the phagemid systems and conventional quorum sensing systems, would be an extremely valuable contribution to the field. The authors have already done some characterization of this nature in Fig. 2e,f for the phagemid system, and I think this work is some of the most compelling work in the current manuscript.

Conclusion:

I would like to re-emphasize that I think the work described in the manuscript is of high quality, and as such I think that it could be published without additional experiments in a more specialized journal. But I believe a paper in *Nature Communications* should provide a significant conceptual or technical advance for the field, which is missing from the current version of this manuscript. However, I believe the authors have the potential to reach that point, perhaps by following some of the directions I outlined above—to this point, I personally think that the head-to-head comparison of the phagemid communication channels versus quorum sensing channels would be the most accessible direction to pursue that would be sufficient to give the work a major significance. The development of alternative intercellular communication systems that can overcome the challenges associated with conventional quorum sensing channels is of great importance to the field. I believe that the intercellular communication field would benefit greatly from such efforts by the authors to extend their current work.

Additional Technical Comments:

Line 57: Additional references to earlier work in this space should be added and briefly discussed (both here and in the Discussion, e.g., ~Line 336, as appropriate). Specific works include:

Ji W, et al. Specific gene repression by CRISPRi system transferred through bacterial conjugation. *ACS synthetic biology*. 2014 Dec 19;3(12):929-31.

Marken JP, Murray RM. Addressable and adaptable intercellular communication via DNA messaging. *Nature Communications*. 2023 Apr 24;14(1):2358.

Fig. 2D: A labeled colorbar for the heatmap values needs to be shown.

Fig. 2E (and related discussion): The data suggest that 100% of receiver cells are infected with phagemid within 30 minutes of coculture, which is surprising. I believe that this result may be due in part to an artifact of the methodology. When the sender-receiver coculture is transferred to selective media, there will still be phagemids in this solution. Receiver cells can therefore continue to receive the phagemid after the coculture is harvested and selection is applied. In the case of the pBR322 phagemid, which is selected by ampicillin, receiver cells under ampicillin selection experience disruptions in cell wall synthesis. This can cause cell death after several replication cycles, but upon infection by the phagemid, these cells will be rescued from the impact of ampicillin. In contrast, the RSF1030 phagemid is selected by gentamicin, which inhibits protein synthesis. In this case, infection by the phagemid does not rescue the receiver cells because the expression of the antibiotic resistance gene requires protein synthesis, which is inhibited. As such, I believe that the data in Fig. S1A is more representative of the true dynamics of phagemid infection than Fig. 2E. The RSF1030 data should therefore be shown and discussed in the main text, and the pBR322 data should either be re-measured with a protein synthesis inhibiting antibiotic or moved to the SI with a description that highlights this difference between the two systems as a potential explanation for the difference in the observed measurements.

Fig. S2E, F: The values in the heatmaps on the left seem indistinguishable from a matrix of 0 values, which is not a productive visualization. The authors should either rescale the colorbar in the left heatmaps, or print the values of the heatmap entries (as is done for the entries along the main diagonal on the heatmaps on the right). To this point, all heatmaps in the manuscript should have their numerical values displayed within each entry, given that there is already sufficient space to do so.

Line 385: The specific rationale for removing the supernatant from the sender cell preculture is not stated—I assume it is to get rid of already-produced phagemid, but this should be described explicitly.

Methods: I could not find a description of how, exactly the GFP fold change values used throughout the paper are calculated. Are they the ratio of the mean GFP fluorescence levels of the receiver cells in the two experimental conditions, or are they ratio of the geometric means, or the medians, or something else?

References

1. Ortiz ME, Endy D. Engineered cell-cell communication via DNA messaging. *Journal of biological engineering*. 2012 Dec;6:1-2.
2. Nielsen AA, Voigt CA. Multi-input CRISPR/C as genetic circuits that interface host regulatory networks. *Molecular systems biology*. 2014 Nov;10(11):763.
3. Bikard D, Barrangou R. Using CRISPR-Cas systems as antimicrobials. *Current opinion in Microbiology*. 2017 Jun 1;37:155-60.
4. Peters JM, et al. Enabling genetic analysis of diverse bacteria with Mobile-CRISPRi. *Nature microbiology*. 2019 Feb;4(2):244-50.

5. Marken JP, Murray RM. Addressable and adaptable intercellular communication via DNA messaging. *Nature Communications*. 2023 Apr 24;14(1):2358.

6. Du P, Zhao H, Zhang H, et al. De novo design of an intercellular signaling toolbox for multi-channel cell–cell communication and biological computation. *Nature communications*. 2020 Aug 24;11(1):4226.

Version 1:

Reviewer comments:

Reviewer #1

(Remarks to the Author)

I would like to thank the authors for answering the majority of my questions - and those of the other reviewer. In particular, I think the addition of the comparison to quorum molecule signalling highlights a significant advantage of this approach.

One answer that I was not satisfied with was the argument against implementing the XOR and XNOR gates. The NOT and NOR composition of your gates, make all logic computations possible but makes some particularly complex. Building an XOR gate from NOT and NOR requires five NOR gates: $\text{NOT}(\text{NOR}(\text{NOR}(A, \text{NOR}(A,B)), \text{NOR}(B, \text{NOR}(A,B))))$. Is this possible with the tools that you currently have and what would the gate quality be? Or are you currently limited in the gates that you can construct. I think it is important for readers to be able to understand the level of complexity that is required for "simple" computations, and the level of complexity that is achievable within a single cell.

This further highlights another consideration - what logic is being performed in which cells, and using which molecular mechanism? In your current implementations, there is some clever off-loading of computation to the sender cells using more standard genetic tools of chemical inducers and transcription factors e.g. the sender cells for the NAND gate perform a NOT of the signal before sending to the receiver. This becomes more extreme for the 4-input gates, where the senders are performing AND logic on the chemical signals before sending to the receivers. I think this is a perfectly valid "hybrid" approach, but it is not really discussed. Could the intra-cellular behaviour of the receivers have been implemented with more typical transcription factor logic (e.g. Cello), and the inter-cellular communication affected through M13 and sgRNAs? If a reader wanted to use your technology, can you help them to understand which components of a computation to put in which cells, and which should be implemented using sgRNAs and which using transcription factors?

Reviewer #2

(Remarks to the Author)

I would like to thank the authors for taking the time and resources to implement my feedback through additional experiments. I believe the manuscript is much strengthened by these additional results, which make important contributions to the field of engineered intercellular communication. I enthusiastically support the publication of this manuscript.

Engineering Intercellular Communication using M13 Phagemid and CRISPR-based Gene Regulation for Multicellular Computing in *Escherichia coli*

Point-by-point response to reviewer comments

Reviewer #1 (Remarks to the Author):

The authors have produced an inducible M13 phage communication system, sending sgRNAs which repress transcription from engineered circuits in receiver cells. This was demonstrated by implementing digital logic bio-computation. The work is presented very clearly and is rather timely given preprints from other groups (highlighted here by the authors).

We thank the reviewer #1 for their positive and constructive feedback on our manuscript. We have now addressed the reviewer's comments. Please find a detailed point-by-point response below.

My only comments are minor details that need to be addressed.

- In figures 3B and 3E, can you provide a control to compare the receiver cell fluorescence without the sender cell present.

We have now included the controls for the receiver cells without the sender cells present in figures 3B and 3E. Please find the updated version of these panels below.

Figure 3. Establishing inducible phagemid transfer in one-input logic gates. **A. Schematic representation of sender and receiver cells for an inducible NOT gate.** Gene VIII (*gVIII*) was deleted on helper plasmid (HP) #1 and added to HP#2 under the control of an inducible promoter. **B. Performance of different inducible promoters regulating P8 production.** Green fluorescence (arbitrary units, a. u.) of the receiver cells were measured after 4 hours of co-culturing with inducible sender cells with the indicated promoters controlling expression of *gVIII* in the presence (green) or absence (grey) of the corresponding inducer: aTc ($0.1 \text{ mg liter}^{-1}$), IPTG (1 mM), AHL (10 mM) and arabinose (1%). The initial sender-to-receiver ratio was 3:1. Data represent the mean \pm SD of 3 independent biological replicates. WT = Wild-type (strain JM101), Δ araBAD: strain JM101 with genes *araBAD* deleted. **C. TEM images of sender cells in the presence or absence of 1 mg liter^{-1} aTc inducer, grown for 4 hours at 37°C with 1000 rpm shaking.** **D. Schematic representation of one version of an inducible BUF / 'Yes' gate.** **E. Performance of two BUF gate variants using inducible phagemid sender cells.** Green fluorescence (a. u.) of the receiver cells was measured after 4 hours of co-culturing with inducible sender cells (aTc-inducible) for BUF gates with sgRNA-2 transduced (as indicated in D). A second variant was tested in which sgRNA-3 is transduced and sgRNA-5 represses sfGFP. The initial sender-to-receiver ratio was 3:1. Data represent the mean \pm SD of 3 independent biological replicates.

- Please report quality of each logic gate as lowest ON state divided by highest OFF state; this has become a useful convention for quickly understanding the efficacy of different bio-computation approaches.

We thank the reviewer for bringing this convention to our attention. We now report the quality of each logic gate accordingly in the current version of figure 4 and 5.

Figure 4. Two-input Boolean logic gates in multicellular consortia **A. 2-input NOR gate.** Left: A schematic representation of an inducible NOR gate design. Right: Average of green fluorescence (a.u.) measured by flow cytometry in the presence or absence of 0.1 mg liter⁻¹ aTc and 1 mM IPTG inducers. Data represents the mean +/- SD of 5 independent biological replicates. **B. 2-input OR gate.** Left: A schematic representation of an inducible 'OR' gate design. Right: Average of green fluorescence (a.u.) measured by flow cytometry in the presence or absence of 0.1 mg liter⁻¹ aTc and 1 mM IPTG inducers. Data represents the mean +/- SD of 6 independent biological replicates. **C. 2-input AND gate.** Left: A schematic representation of an inducible 'AND' gate design. Right: Average of green fluorescence (a.u.) measured by flow cytometry in the presence or absence of 0.025 mg liter⁻¹ aTc and 0.25 mM IPTG inducers. Data represents the mean +/- SD of 3 independent biological replicates and error bars represents standard deviation. **D. 2-input NAND gate.** Left: A schematic representation of an inducible 'NAND' gate design. Right: Average of green fluorescence (a.u.) measured by flow cytometry in the presence or absence of 0.1 mg liter⁻¹ aTc and 1 mM IPTG inducers. Data represents the mean +/- SD of 6 independent biological replicates. For all 2-input gates, the initial sender-to-receiver ratio was 3:1. Fold-change (FC) is calculated by dividing the 'ON' state by the 'OFF' state, while the quality score (Q) was calculated by dividing the lowest 'ON' state by the highest 'OFF' state.

- There are a few missing 2-input gates which I think should be included for completeness. Yes, indeed we are missing some 2-input gates, e.g., the XOR and XNOR gates. Our primary goal in this manuscript was to establish intercellular communication for distributed biocomputing using phagemids. We believe that the 2-input and 4-input gates we have constructed successfully demonstrate our ability to transmit signals, as well as to induce and inhibit phagemid transfer. As Reviewer 2 correctly pointed out, our 2-input gates do not achieve anything that would be impossible through other means. Therefore, rather than constructing the remaining 2-input gates, we believe our efforts are better directed toward a project that fully leverages multicellular computing to enable more complex computations, such as addition or multiplication of numbers, surpassing what has been achieved in monocultures. We hope the reviewer understands our decision on how to best allocate our limited resources.

- Were the two 4-input gates the only ones that were attempted? Why were those particular gates chosen?

Yes, we only attempted the 4-input AND and NAND gates. Using the 2-input gates we had already built, we successfully expanded them into 4-input gates without modifying the circuit in the receiver cells. This contrasts with previously published 4-input gates, where converting a 2-input gate into a 4-input gate typically required optimization or even an additional directed evolution step to ensure proper functionality (Ref 7).

As mentioned in our response to the previous point, we believe we could construct additional 2- and also 4-input gates. However, we consider the ones we have already built sufficient as a proof of principle and now prefer to focus on enabling more complex computations.

- Check references – 7 and 9 are the same

Thanks for pointing out this mistake. We have now removed the duplicated reference.

- Line 329 “optimzse” -> optimize

Corrected.

- Please include catalogue numbers for chemicals used in the methods – this is crucial for reproducibility.

We have now added the catalogue numbers in the methods:

Line 418: When required, gentamicin (25 mg liter⁻¹) (Sigma-Aldrich cat. no. G1914), ampicillin (100 mg liter⁻¹) (Sigma-Aldrich cat. no. A9518), kanamycin (50 mg liter⁻¹) (Sigma-Aldrich cat. no. K4000), spectinomycin (50 mg liter⁻¹) (Sigma-Aldrich cat. no. S4014), arabinose (1% w/v) (Sigma-Aldrich cat. no. A3256), anhydrotetracycline (aTc) (0.1 mg liter⁻¹) (Chemodex cat. no. 13803-65-1), N-(β -Ketocaproyl)-L-homoserine lactone (AHL/OC6-HSL) (10 μ M) (Sigma-Aldrich cat. no. K3007), N-(3-Oxotetradecanoyl)-L-homoserine lactone (AHL/OC14-HSL) (10 μ M) (Sigma-Aldrich cat. no. O9264), and Isopropyl β -D-1-thiogalactopyranoside (IPTG) (1 mM) (Sigma-Aldrich cat. no. PHG0010) were added to the media.

- What did you use to cover the 96-well plates in the plate reader experiments?

For incubation in deep-well plates (phagemid transfer assay), we used Breathe-Easier sealing membranes, while for measurements in 200 μ L 96-well plates using the plate reader, we used the original lid supplied with the plates. We have now included these details in the Methods section:

Line 465: The 96x deep well plate was then covered by Breathe-Easier sealing membrane (Merck, cat. no. Z763624).

Line 491: We added 120 μ L of 0.05 OD₆₀₀ bacterial suspensions per well in a 96-well CytoOne plate (Starlab) and supplemented it with inducers to reach the desired concentrations. The plate was covered with the supplied lid and were incubated at 37°C with double-orbital shaking in a Synergy H1 microplate reader (Biotek) running Gen5 3.04 software.

- The fluorescence normalisation carried assumes in this manuscript assumes a linear relationship between autofluorescence and absorbance, which is not true in the GFP spectrum. See <https://pubs.acs.org/doi/full/10.1021/acssynbio.0c00296>

We would like to clarify that the fluorescence measurements for our phagemid transfer assays were performed using flow cytometry and therefore we did not measure any absorbance. Plate reader analysis was only performed for the measurement of promoter strengths using sfGFP as a proxy (Figure S2B and Figure S2C). Nevertheless, we have now adapted our data analysis according to the guidelines described in the publication of Fedorec et al. In particular, we subtracted now a non-fluorescent control from our samples. Therefore, the figures and the reported fold-changes changed slightly. We did not

convert the arbitrary units into absolute units, as we are mainly interested in fold-changes, but we regularly calibrate our flow cytometers with fluorescent beads.

We have now better described our data analysis method for both flow cytometry and plate reader assays:

Line 474: First, to discriminate between cells and other particles, all measured events were gated by forward scatter height (FSC.H) > 1000 arbitrary unit (a.u.) and side scatter height (SSC.H) > 200 a.u. Second, we excluded doublets by plotting the FSC.H against the forward scatter area (FSC.A) and set a gate for the events with approximately 1:1 ratio of FSC.H to FSC.A. We recorded 30,000 events of singlet cells. We then set a gate to select for the receiver cells ('sender/receiver threshold' in Figure 2C) for red fluorescence (PE.Texas.Red.H) above 2000 a.u. Finally, the data were analysed and visualized in R using RStudio 1.4.1106 (R 3.4.0). The reported GFP fluorescence values were calculated as the geometric mean of individual biological replicates. Then we subtracted the basal green fluorescence value of the cells, which was measured from the non-fluorescence JM101 control. GFP reporter fold change (FC, Figure 4 & 5) was then determined by dividing the mean GFP fluorescence levels of the intended 'ON' state by those of the 'OFF' state. For calculating the fraction of receiver cell population that received the phagemid messages (Figure 2D), we set additional gates for the 'ON' state and 'OFF' states by setting a threshold for green fluorescence (FITC.H) above or below 10,000 a.u., respectively (Figure S12).

Line 495: The fluorescence levels were processed as follows: (i) subtracting the fluorescence signal of a blank sample, (ii) dividing the resulting value by the absorbance at 600 nm to account for differences in bacterial concentration, and (iii) subtracting the bacterial autofluorescence of a strain without reporter genes. The normalized data were plotted using ggplot2 package in RStudio 1.4.1106 (running R 3.4.0).

Reviewer #2 (Remarks to the Author):

In this work, Kusumawardhani, Zoppi, Avendaño, and Schaerli expand on Ortiz and Endy's earlier implementation of M13 phagemid-based intercellular communication [Ref. 1] by introducing two innovations: (1) allowing the inducible production of the phagemid by sender cells, and (2) coupling the phagemids to CRISPRi logic circuits in the receiver cells.

The manuscript is well written and the experimental work is well executed and communicated clearly. However, while it has the potential to do so, I feel that the current version of the manuscript does not provide the field with results of sufficient significance to justify publication in a broad-audience journal like Nature Communications. Below, I will describe my thoughts about how the authors might extend their current work to achieve this significance.

We thank the reviewer #2 for their valuable and constructive feedback. We have now implemented several of their suggestions, which we believe have significantly improved the impact and significance of our manuscript. We hope the reviewer agrees.

On the significance of the work:

My fundamental concern is that while the paper is focused on expanding the capabilities of DNA-based intercellular communication systems, most of the experiments presented in the paper do not engage in a meaningful way with either intercellular communication or distributed computation. The addition of chemical inducibility to Ortiz and Endy's M13 system is a legitimate advance that has utility for future efforts in engineering multicellular consortia, but this is not the case for the construction and characterization of CRISPRi logic circuits, which constitute the bulk of the work. This can be illustrated by noting that all of the circuits in Figs 3-5 follow the same overall architecture:

Phagemid-based Circuit Architecture (this work):

1. The sender cells contain an inducible M13 channel and a constitutive sgRNA on the phagemid.
2. The receiver cell contains a latent CRISPRi circuit that takes the transferred sgRNAs as inputs.

The above architecture is logically identical to the following single-strain implementation:

Single-strain Circuit Architecture:

1. One cell strain contains the inducible sgRNA inputs, as well as the latent CRISPRi circuit that takes those sgRNAs as inputs.

There is no clear reason why one should choose to pick the multi-strain implementation over the single-strain implementation when implementing a circuit.

We agree with the reviewer that the submitted manuscript did not demonstrate a function that would be impossible to implement in monocultures. Indeed, we had acknowledged that already in the discussion: "In the circuits presented in this paper, all messages end up in the same receiver cell. This allowed us to establish and optimize the combination of M13 phagemid-based intercellular communication with gene regulation using CRISPRi. It allowed us to build four-input AND and NAND gates matching the complexity of logic gates built so far in monoculture".

However, our approach enabled significantly faster construction of 4-input gates compared to their implementation in single cells. More importantly, we developed and demonstrated a methodology that lays the foundation for performing more complex computations, such as addition or multiplication of numbers—capabilities that go beyond what has been achieved in monocultures. We view this as a

significant milestone worth sharing with the community now, especially as we make available over 40 plasmids on Addgene, even as we continue progressing toward the next milestone. That said, we have now incorporated two of your suggestions to further enhance the impact of this manuscript. Please see below for details.

Even if one insists on distributing the circuit between multiple cell types, there is no clearly presented benefit of using phagemids for intercellular communication as opposed to a logically equivalent implementation using conventional quorum sensing systems:

Quorum sensing-based Circuit Architecture:

1. The sender cells contains an inducible AHL synthase.
2. The receiver cell contains a latent CRISPRi circuit, as well as cassettes that inducibly express the sgRNA inputs for the CRISPRi circuit based on the presence of the cognate AHL.

Convincingly presenting the benefits of the phagemid-based multicellular implementation, over conventional multicellular implementations and/or over single-strain implementations, will be the central challenge of the authors in elevating their current work to a higher level of significance. This needs to be accomplished both through revisions on the narrative of the paper as well as the choice of experiments that are presented.

On comparing multi-strain implementations to single-strain implementations:

The ability to construct complex CRISPRi logic circuits in single-strain implementations is now well-established in the field [Ref. 2]. Transferring CRISPR components intercellularly to conduct gene editing or DNA cleavage in the recipient cells is also routinely done [Ref. 3], even in the context of CRISPRi [Ref. 4]. It is therefore unsurprising that a multicellular CRISPRi logic circuit, distributed in the way that is presented in the paper, would work.

The authors must therefore demonstrate that their system is able to create circuits that can achieve qualitatively different functionalities from conventional single-strain CRISPRi implementations.

For example, what if one were interested in building circuits that don't sense the concentrations of target inducer molecules, but rather the densities of target cell strains within a population? The receiver cell could then act as a dynamic reporter of the population composition of a multistrain consortium, with different logic gates used to give readouts of different population states. In such an implementation, each monitored cell could constitutively express phagemids containing a unique sgRNA input into the logic circuit. Building such a circuit would require additional characterization experiments that assess, e.g., the sensitivity of M13 detection (i.e., in a sender-receiver coculture, what is the threshold density of sender cells below which receivers can no longer reliably detect M13-driven communication?), but would constitute a type of circuit that takes advantage of the unique properties of multi-strain implementations.

We thank the reviewer for this suggestion. We have now implemented this idea and applied the (constitutive) phagemid transfer system for detecting the presence of sender cells using our circuit. This new data is now added to Figure 2, panel G (See below). We have added the following texts in the result and discussion section regarding this experiment:

Line 161: As an application of this NOT gate circuit, we built a genetic circuit capable of the presence of the sender cells in a co-culture. This circuit is based on two independent NOT gates (pHK001.23), where sgRNA-2 represses sfGFP and sgRNA-3 represses mCherry allowing the detection of two distinct

sender cell populations independently (Figure 2G). We tested the sensitivity of this circuit by co-culturing receiver cells at OD_{600} 0.5 ($3.2 \pm 0.375 \times 10^5$ cfu) with sender cells sending sgRNA-2 at different densities (Figure 2G). We found that this circuit could reliably detect sender cells at concentrations approximately 1000 times lower than receiver cells (approximately 300 sender cells among 300'000 receiver cells) within 4 hours of co-culturing. Notably, this high sensitivity remained unaffected even in the presence of a second sender cell population expressing a different sgRNA (sgRNA-3) at OD_{600} 0.5.

Line 345: We also leveraged the phagemid transfer system to detect the presence of highly diluted sender cells in a co-culture (Figure 2G).

Figure 2G. Measurement of the sender cell density using pET/pBR322-based phagemids. Receiver cells harbored two independent NOT gates allowing the detection of two distinct sender cell populations independently. The sensitivity of this circuit was tested by co-culturing receiver cells at OD_{600} 0.5 ($3.2 \pm 0.375 \times 10^5$ cfu) with only sender cells sending sgRNA-2 at different densities or challenged by an additional sender cell population at OD_{600} 0.5 sending sgRNA-3. Data in G represent the mean \pm SD of 4 independent biological replicates.

On comparing phagemid-based communication to quorum sensing channels:

Ortiz and Endy highlighted the specific advantages of DNA-based communication over other forms of intercellular communication as (1) Message-channel decoupling, (2) Flexibility of encoded message, and (3) Specificity of message receipt. Marken and Murray [Ref. 5] also recently identified (4) Large information capacity of the message and (5) Dynamic mutability of the message as additional unique advantages of DNA-based communication. With the exception of message-channel decoupling, which is a structural property of the M13 phagemid system, none of the other unique advantages of DNA-based communication are leveraged in this work.

The authors do make the point that by transferring sgRNAs along phagemid channels, one can reach a total number of communication channels that is dictated by the (potentially very large) number of orthogonal sgRNA-binding site pairs, and empirically demonstrate 6 such pairs. However, the ability to reach up to 8 orthogonal intercellular channels has already been demonstrated without DNA-based communication [Ref. 6]. If the authors want to lean into the number of possible communication channels as a central advance of their work, they should empirically demonstrate the ability to scale far beyond the current non-DNA record of 8 orthogonal channels. There are many claims in the literature about the scalability of sgRNA-based systems, but if the authors were able to empirically demonstrate a dramatic increase in the number of simultaneous orthogonal communication channels this way, that would certainly constitute a major advance in the field.

So far, we have not yet implemented eight different sgRNAs and their corresponding binding sites into this phagemid system. However, given the availability of computational tools for sgRNA design (refs 57, 58) and the successful construction of arrays containing up to 24 gRNAs (ref 59), we do not anticipate this being a major limitation. The greater challenge lies in ensuring that a single cell can receive all eight signals. If each plasmid encodes only one sgRNA, this would require the uptake of eight

different plasmids with orthogonal origins of replication, which we consider unlikely to be feasible. However, it is worth noting that in Ref 6, no single cell type receives more than two signals simultaneously. In this regard, we see no fundamental reason why our system could not match the capabilities of the system published in Ref 6. That said, we acknowledge that we have not yet had the time to experimentally validate this expansion. We now explicitly mention the potential for scaling up to more sgRNAs in the discussion:

Line 399: Given the availability of sgRNA design tools^{57,58} and the successful use of up to 24 orthogonal and multiplexed sgRNAs⁵⁹, we anticipate that our method holds significant potential for scaling up to more complex biocomputing systems.

An alternative approach is to conduct head-to-head characterizations of signaling performance between quorum sensing-based systems and the phagemid system, which to my knowledge has not been done. Characterizing communication-relevant properties such as the sensitivity of the receiver cells to the sender cell density (as I described above), the timescale from the onset of signal transmission to receiver response, etc., and how they vary between the phagemid systems and conventional quorum sensing systems, would be an extremely valuable contribution to the field. The authors have already done some characterization of this nature in Fig. 2e,f for the phagemid system, and I think this work is some of the most compelling work in the current manuscript.

Thank you for this suggestion. We have implemented this idea by constructing a new circuit (pHK001.3QS), a NOT gate where repression is mediated by sgRNA-3 (see schematic in Figure 3F below). The mCherry reporter can be repressed either by the transfer of a phagemid encoding sgRNA-3 or by inducing sgRNA-3 expression under the control of a quorum-sensing (QS) system, specifically via cinR - OC14-HSL at the Pcin promoter.

We first confirmed that adding 10 μ M OC14-HSL to the medium or coculturing with sender cells constitutively producing the phagemid led to a comparable decrease in mCherry fluorescence (Fig. S3G). Next, we induced either phagemid transfer or production of OC14-HSL production (via cinI) in sender cells. The signal transmission via phagemid was significantly faster and stronger than via QS. We attribute this difference to the dilution of the secreted QS molecule in liquid culture, whereas a single successfully transferred phagemid is sufficient to produce a robust response in the receiver cell. Therefore, while QS-based intercellular communication systems have been shown to work effectively on agar plates, as demonstrated by Du et al. and many others, liquid culture experiments with QS-based communication usually rely on conditioned media or strategies to enrich for signal accumulation. Instead, our results highlight that phagemid-based communication ensures a highly sensitive response in liquid culture. We have incorporated this finding into the results and discussion sections of the manuscript:

Line 210: Next, we sought to compare the performance of our phagemid-based intercellular communication to the well-established quorum-sensing-based communication in liquid culture. To do this, we designed a new circuit (pHK001.3QS), which is based on a NOT gate where repression is mediated by sgRNA-3 (Figure 3F). This circuit contains a sgRNA-3 under the regulation of the CinR/Pcin system, which is OC14-HSL inducible. We selected this quorum-sensing system for its good dynamic range and tight control^{44,45}. The repression of mCherry can thus be triggered by the presence of a phagemid encoding sgRNA-3 or by the presence of OC14-HSL in the medium. As shown in Figure S3-G, the presence of sender cells constitutively producing phagemids resulted in a comparable repression of mCherry to that achieved by adding 10 μ M OC14-HSL to the medium.

After confirming that the circuit functioned correctly for both modes of communication, we compared their efficiency using two different inducible sender cells: one in which M13 phagemid transfer was inducible via IPTG (Ptac-gVIII/pHK326) and another in which IPTG induced cinI expression (pHK-cinI), leading to OC14-HSL production. We monitored the repression of mCherry over time following induction at the start of co-culturing. While phagemid transfer led to mCherry repression

within 3-4 hours of co-culturing, quorum-sensing-based cell-cell communication required overnight co-culturing to achieve just a two-fold repression, which is much lower than the response observed with the phagemid-based communication system (Figure 3F).

Line 356: We also demonstrated that in liquid culture, phagemid-based communication led to a faster and stronger response than quorum-sensing-based communication in a comparable setup (Figure 3F). We attribute this difference between the two systems to the dilution of the secreted quorum-sensing molecules in liquid culture, whereas a single successfully transferred phagemid is sufficient to produce a robust response in the receiver cell. This aligns with the high sensitivity of phagemid detection demonstrated in Figure 2G. While quorum-sensing-based intercellular communication systems have been shown to work effectively between close-by colonies on agar plates^{28,50} or on conditioned media for receiver activation^{51,52}, our results highlight that phagemid-based communication constitutes a highly efficient intercellular signalling mechanism in liquid culture.

F. Comparison between quorum sensing and phagemid-based messaging

Figure 3F. Comparison of phagemid-based to quorum-sensing-based intercellular communication in liquid culture. Receiver cells contained a mCherry reporter that can be repressed by sgRNA-3. This sgRNA-3 is produced if the receiver cells sense OC14-HSL or if they receive a message phagemid containing sgRNA-3. The inducible phagemid sender cells contained P_{tac} regulated gVIII, whereas the quorum-sensing sender cells contained P_{tac} regulated cinI for OC14-HSL production. Red fluorescence (a. u.) of the receiver cells was monitored at 1, 2, 3, 4, 5, 6, and 24 hours of co-culturing (sender-to-receiver ratio 3:1) with inducible sender cells (IPTG-inducible phagemid transfer or OC14-HSL production). mCherry-fold repressions were calculated from these measurements. Data represent the mean \pm SD of 4 independent biological replicates.

Conclusion:

I would like to re-emphasize that I think the work described in the manuscript is of high quality, and as such I think that it could be published without additional experiments in a more specialized journal. But I believe a paper in Nature Communications should provide a significant conceptual or technical advance for the field, which is missing from the current version of this manuscript. However, I believe the authors have the potential to reach that point, perhaps by following some of the directions I outlined above— to this point, I personally think that the head-to-head comparison of the phagemid communication channels versus quorum sensing channels would be the most accessible direction to pursue that would be sufficient to give the work a major significance. The development of alternative intercellular communication systems that can overcome the challenges associated with conventional quorum sensing channels is of great importance to the field. I believe that the intercellular communication field would benefit greatly from such efforts by the authors to extend their current work.

We thank the reviewer for recognizing the high quality of our work and hope that the additional experiments - including sender cell quantification and the direct comparison between phagemid-based and quorum-sensing communication - enhanced the impact of our manuscript.

Additional Technical Comments:

Line 57: Additional references to earlier work in this space should be added and briefly discussed (both here and in the Discussion, e.g., ~Line 336, as appropriate). Specific works include:

Ji W, et al. Specific gene repression by CRISPRi system transferred through bacterial conjugation. ACS synthetic biology. 2014 Dec 19;3(12):929-31.

Marken JP, Murray RM. Addressable and adaptable intercellular communication via DNA messaging. Nature Communications. 2023 Apr 24;14(1):2358.

We have added the suggested papers in the introduction and discussion sections.

Line 57: Marken and Murray developed an addressable and adaptable framework for DNA-based communication in *E. coli*, by leveraging plasmid conjugation³⁰. In a separate study, Weiyue et al. employed bacterial conjugation to deliver an inducible CRISPRi system, repressing the mRFP gene in a target *E. coli* strain³¹.

Line 404: (references added)

Fig. 2D: A labeled colorbar for the heatmap values needs to be shown.

Labeled colorbars for the heatmap values have been added for figure 2D. See below the current version of this figure panel.

Figure 2D. Orthogonality assay of different sgRNAs in sender pBRR322-based phagemids (pHK302.y, ampicillin resistance) (y axis) and binding sites in receiver cells (pHK001.x) (x axis). The initial sender to receiver ratio was 1:1. Left: Mean +/- SD of 3 independent biological replicate of sfgFP repression fold-change determined by flow cytometry. Right: fraction of receiver cells with fluorescent below the chosen threshold (shown in C). Data represent the mean of 3 independent biological replicates.

Fig. 2E (and related discussion): The data suggest that 100% of receiver cells are infected with phagemid within 30 minutes of coculture, which is surprising. I believe that this result may be due in part to an artifact of the methodology. When the sender-receiver coculture is transferred to selective media, there will still be phagemids in this solution. Receiver cells can therefore continue to receive the phagemid after the coculture is harvested and selection is applied. In the case of the pBRR322 phagemid, which is selected by ampicillin, receiver cells under ampicillin selection experience disruptions in cell wall synthesis. This can cause cell death after several replication cycles, but upon infection by the phagemid, these cells will be rescued from the impact of ampicillin. In contrast, the RSF1030 phagemid is selected by gentamicin, which inhibits protein synthesis. In this case, infection by the phagemid does not rescue the receiver cells because the expression of the antibiotic resistance gene requires protein synthesis, which is inhibited. As such, I believe that the data in Fig. S1A is more

representative of the true dynamics of phagemid infection than Fig. 2E. The RSF1030 data should therefore be shown and discussed in the main text, and the pBR322 data should either be re-measured with a protein synthesis inhibiting antibiotic or moved to the SI with a description that highlights this difference between the two systems as a potential explanation for the difference in the observed measurements.

We thank the reviewer for pointing this out. We have changed the resistance and re-measured the pBR322-based phagemid transfer using gentamicin resistance selection instead of ampicillin. As correctly predicted by the reviewer, we observed that pBR322 transfer trends are now more similar to RSF1030 (Figure S1A). We added the following text in the main text and replaced Figure 2 (panel E and F) and Figure S1 with the updated data.

Line 119: Next, we quantified the dynamics of phagemid transduction between sender and receiver cells. Using a 2:1 sender-to-receiver cell ratio (see justification in the next paragraph), we took samples at 0.5, 1, 2, 4, and 6 hours. We then plated the cultures on two types of agar plates: one with antibiotics allowing growth of all receiver cells and another that selectively allowed growth of only those with the phagemid. We counted the colony-forming units on each plate. In this experiment, both message phagemids contained a gentamicin resistance gene to ensure comparability. We observed phagemid transduction occurring within 2 hours for both pBR322 and RSF1030 phagemids (Figure 2E and S1A). Additionally, we also followed sfGFP repression at these timepoints. It took around 4 hours to record a 20-fold sfGFP repression for both constructs (Figure 2F and Figure S1B).

Furthermore, we also compared the effect of different antibiotic pressures on the pBR322 phagemid, namely gentamicin versus ampicillin (Figure S1 C & D). To achieve a level of repression comparable to that of the ampicillin-selected phagemid with a 1:1 sender-to-receiver cell ratio, the ratio had to be increased to 2:1 for the gentamicin-selected phagemid. Even with this adjustment, the number of colonies growing after transduction during the first hour was lower with the gentamicin-selected phagemid (Figure S1C), while the sfGFP- fold repression was comparable to that of the ampicillin-selected phagemid (Figure S1D). We attribute these differences to the distinct mechanisms of action of the two antibiotics⁴⁰. Under ampicillin selection, receiver cells experience disruptions in cell wall synthesis, which can lead to cell death after several replication cycles. Phagemid infection rescues these cells by providing antibiotic resistance, allowing them to survive despite ampicillin's effects. In contrast, gentamicin inhibits protein synthesis, making it more difficult for phagemid infection to rescue receiver cells, as the expression of the antibiotic resistance gene depends on protein synthesis.

Figure 2E. Transduction of pBR322-based message phagemid (with gentamicin resistance) over time determined by calculating colony forming units of successfully transduced cells using selective plating divided by total amount of receiver colonies. The initial sender-to receiver-ratio was 2:1. Corresponding data of pBR322 with ampicillin resistance and for the RSF1030 backbone can be found in Fig. S1. **F. Fold-change of sfGFP reporter repression over time for pBR322-based phagemid.** Data in E and F represent the mean +/- SD of 4 independent biological replicates.

Figure S1. Characterization of phagemid transfer. **A.** Transduction of the pBR322 phagemid compared to the RSF1030 message phagemid (both with gentamicin resistance) over time. The initial sender to receiver ratio was 2:1. Transduction rates were determined by counting colony-forming units of successfully transduced cells on selective plates divided by the total amount of receiver colonies. **B.** Fold-change of sfGFP reporter repression over time for pBR322 and RSF1030-based phagemids. **C.** Transduction of pBR322-based message phagemid with gentamicin (sender-to-receiver ratio 2:1) or ampicillin resistances (sender-to-receiver 1:1) overtime. **D.** Fold-change of sfGFP reporter repression over time for pBR322-based message phagemid selected by gentamicin resistance or by ampicillin resistance. **E.** Fold-change of sfGFP reporter repression for pBR322-based phagemid with promoter J23119 regulating the expression of sgRNA-1, sgRNA-2, or sgRNA-3. Data in A, B, C, D, and E represent the mean \pm SD of 4 independent biological replicates.

Fig. S2E, F: The values in the heatmaps on the left seem indistinguishable from a matrix of 0 values, which is not a productive visualization. The authors should either rescale the colorbar in the left heatmaps, or print the values of the heatmap entries (as is done for the entries along the main diagonal on the heatmaps on the right). To this point, all heatmaps in the manuscript should have their numerical values displayed within each entry, given that there is already sufficient space to do so.

We have now added numerical values for each entry in figures S2E & F. See below the current version of these figure panels.

Figure S2 E. Orthogonality assay of aTc-inducible phagemid transfer for NOT gates with different sgRNAs and binding sites pairs in the presence and absence of aTc. **F.** Orthogonality assay of aTc-inducible phagemid transfer for BUF / 'Yes' gates with different sgRNAs and binding sites pairs in the presence and absence of aTc. In this receiver cells's circuit, we placed the sfGFP reporter gene downstream of a promoter J23100 and a binding site for sgRNA-4 (pHK506.4) or sgRNA-5 (pHK506.5). The production of sgRNA-4 (pHK506.4) or sgRNA-5 (pHK506.5) can in turn be repressed by message phagemids containing sgRNA-2 (pHK302.2) or sgRNA-3 (pHK302.3), respectively. Data in E and F represent the mean \pm SD of 3 independent biological replicates.

Line 385: The specific rationale for removing the supernatant from the sender cell preculture is not stated— I assume it is to get rid of already-produced phagemid, but this should be described explicitly.

We did this to remove the antibiotics added to select for the phagemid during the overnight growth and refreshed culture of the sender cells since we do not want to apply selection pressure for the receiver cells with message phagemids. However, you are correct that already produced phagemids are also removed in this step. We have now clarified this in the methods:

Line 456: For sender cells, we pelleted the cells with centrifugation and removed the supernatant. We resuspended the cell pellets with 2x YT containing kanamycin 50 mg liter⁻¹ (for constitutive phagemid transfer) or 2x YT containing kanamycin 50 mg liter⁻¹ and spectinomycin 50 mg liter⁻¹ (for inducible phagemid transfer) and adjusted samples to an OD600 ~ 0.5, thus removing already produced phagemids and ensuring that the experiment will be carried without selection pressure for message transmission.

Methods: I could not find a description of how, exactly the GFP fold change values used throughout the paper are calculated. Are they the ratio of the mean GFP fluorescence levels of the receiver cells

in the two experimental conditions, or are they ratio of the geometric means, or the medians, or something else?

We have added the following explanation in the Methods section.

Line 479: The reported GFP fluorescence values were calculated as the geometric mean of individual biological replicates. Then we subtracted the basal green fluorescence value of the cells, which was measured from the non-fluorescence JM101 control. GFP reporter fold change (FC, Figure 4 & 5) was then determined by dividing the mean GFP fluorescence levels of the intended 'ON' state by those of the 'OFF' state.

References

1. Ortiz ME, Endy D. Engineered cell-cell communication via DNA messaging. *Journal of biological engineering*. 2012 Dec;6:1-2.
2. Nielsen AA, Voigt CA. Multi-input CRISPR/C as genetic circuits that interface host regulatory networks. *Molecular systems biology*. 2014 Nov;10(11):763.
3. Bikard D, Barrangou R. Using CRISPR-Cas systems as antimicrobials. *Current opinion in Microbiology*. 2017 Jun 1;37:155-60.
4. Peters JM, et al. Enabling genetic analysis of diverse bacteria with Mobile-CRISPRi. *Nature microbiology*. 2019 Feb;4(2):244-50.
5. Marken JP, Murray RM. Addressable and adaptable intercellular communication via DNA messaging. *Nature Communications*. 2023 Apr 24;14(1):2358.
6. Du P, Zhao H, Zhang H, et al. De novo design of an intercellular signaling toolbox for multi-channel cell-cell communication and biological computation. *Nature communications*. 2020 Aug 24;11(1):4226.

REVIEWERS' COMMENTS

Reviewer #1 (Remarks to the Author):

I would like to thank the authors for answering the majority of my questions - and those of the other reviewer. In particular, I think the addition of the comparison to quorum molecule signalling highlights a significant advantage of this approach.

One answer that I was not satisfied with was the argument against implementing the XOR and XNOR gates. The NOT and NOR composition of your gates, make all logic computations possible but makes some particularly complex. Building an XOR gate from NOT and NOR requires five NOR gates: $\text{NOT}(\text{NOR}(\text{NOR}(A, \text{NOR}(A,B)), \text{NOR}(B, \text{NOR}(A,B))))$. Is this possible with the tools that you currently have and what would the gate quality be? Or are you currently limited in the gates that you can construct. I think it is important for readers to be able to understand the level of complexity that is required for "simple" computations, and the level of complexity that is achievable within a single cell.

We apologise that we did not answer this question satisfactorily in our previous revision. We now provide designs of the XOR and XNOR gates in supporting Figure S13. Thanks to our modular setup, their designs are very similar to the already implemented 4-input NAND gate and we expect them to perform similarly. We comment now on this in the discussion:

“Although we have not yet implemented the full set of logic gates using our phagemid-based intercellular communication, we have designed two more two-input gates, namely NOR and XNOR gates (Figure S13). In these designs, the sender cells closely resemble those used in the four-input NAND gate, while the receiver cells are identical to those in the NOR and OR gates, respectively.”

Figure S13. Proposed design of XOR and XNOR gates. The sender cells require only minimal modifications compared to those in the four-input NAND gate. The receiver cells are identical to those used in the two-input OR gate (pHK507) for the XOR gate and the NOR gate (pHK009) for the XNOR gate. Specifically, in the sender cells inducers control *cl* and gene VIII expression, where *cl* represses gene VIII. In a sender cell carrying sgRNA2, gene VIII expression is inducible by IPTG via the Ptac promoter, while *cl* expression is induced by aTc via the Ptet promoter. As a result, IPTG alone triggers phagemid production carrying sgRNA2, whereas the presence of both IPTG and aTc inhibits phage particle formation due to *cl*-mediated repression of gene VIII. A similar but inverse regulatory mechanism applies to the sender cell carrying sgRNA3. Here, aTc induces gene VIII expression and the corresponding phagemid, while the simultaneous presence of IPTG and aTc inhibits phage production. Thus, when the two sender cells are co-cultured with receiver cells, phagemid particles are produced only when either IPTG or aTc is present, but not when both are added simultaneously.

This further highlights another consideration - what logic is being performed in which cells, and using which molecular mechanism? In your current implementations, there is some clever off-loading of computation to the sender cells using more standard genetic tools of chemical inducers and transcription factors e.g. the sender cells for the NAND gate perform a NOT of the signal before sending to the receiver. This becomes more extreme for the 4-input gates, where the senders are performing AND logic on the chemical signals before sending to the receivers. I think this is a perfectly valid “hybrid” approach, but it is not really discussed. Could the intra-cellular behaviour of the receivers have been implemented with more typical transcription factor logic (e.g. Cello), and the inter-cellular communication affected through M13 and sgRNAs? If a reader wanted to use your technology, can you help them to understand

which components of a computation to put in which cells, and which should be implemented using sgRNAs and which using transcription factors?

Thank you for highlighting this important point. We have now added following paragraph in the discussion:

“In our designs, we have exclusively used CRISPRi-based gene regulation in receiver cells while leveraging well-established transcription factors to control inducible promoters and invert signals in sender cells. This approach allowed us to avoid expressing dCas9 in sender cells while utilizing dCas9 in receiver cells, where it was already required to process the transferred message. However, we believe that our framework of phagemid-delivered sgRNA messages is highly flexible and can integrate both CRISPR-based gene regulation and traditional transcription factor logic—such as that used in the Cello framework⁸—in both sender and receiver cells. Additionally, carrying out part of the computation in receiver cells with transcription factors could help mitigate the issue of dCas9 competition mentioned earlier.”

Reviewer #2 (Remarks to the Author):

I would like to thank the authors for taking the time and resources to implement my feedback through additional experiments. I believe the manuscript is much strengthened by these additional results, which make important contributions to the field of engineered intercellular communication. I enthusiastically support the publication of this manuscript.

We thank both reviewers for their feedback, which helped us to improve the manuscript.